



# Methane vertical profiles over the Indian subcontinent derived from the GOSAT/TANSO-FTS thermal infrared sensor

Dmitry A. Belikov[1], Naoko Saitoh[1], Prabir K. Patra[2,1], and Naveen Chandra[2]

[1]Center for Environmental Remote Sensing, Chiba University, Chiba, 263-8522, Japan

[2]Research Institute for Global Change (RIGC), Japan Agency for Marine-Earth Science and Technology (JAMSTEC), Yokohama, 236-0001, Japan

*Correspondence to*: Dmitry A. Belikov (d.belikov@chiba-u.jp)

**Abstract**. We examined CH$_4$ variability over different regions of India and the surrounding oceanic regions derived from

thermal infrared (TIR) band observations by the Thermal And Near-infrared Sensor for carbon Observation-Fourier Transform Spectrometer (TANSO-FTS) onboard the Greenhouse gases Observation SATellite (GOSAT) and simulated by the updated MIROC4.0-based Atmospheric Chemistry Tracer Model (MIROC4-ACTM) for the period 2009-2014. This study attempts to understand the sensitivity of the vertical profile retrievals at different layers of the troposphere and lower stratosphere, arising from the averaging kernels and a priori assumptions. We stress that this is of particular importance when the satellite derived

products are analysed using a different ACTMs from that is used as retrieval a priori. A comparison of modeled and retrieved CH$_4$ vertical profiles shows the 22 vertical levels of GOSAT/TANSO-FTS TIR retrievals provide critical information about transport from the top of the boundary layer to upper troposphere and lower stratosphere in a consistent manner. The mean model-GOSAT TIR CH$_4$ mismatch is within 50 ppb, excepting 150 hPa and upward, where the sensitivity of GOSAT/TANSO-FTS TIR observations becomes very low. Convolution of the modeled profiles with GOSAT/TANSO-FTS TIR averaging

kernels reduce the mismatch to below uncertainty. Distinct seasonal variations of CH$_4$ have been observed at the upper atmospheric boundary layer (800 hPa), free troposphere (500 hPa), and upper troposphere (200 hPa) levels over northern and southern regions of India corresponding to the southwesterly monsoon (July–September) and post-monsoon (October–December) seasons. Analysis of the transport and emission contributions to CH$_4$ suggests that the CH$_4$ seasonal cycle over the Indian subcontinent is governed by both the heterogeneous distributions of surface emissions and the influence of the global

monsoon divergent wind circulations. GOSAT/TANSO-FTS TIR observations provide additional information about CH$_4$ observations in this region compared to what is known from in situ data, which is important for improving the accuracy of emission flux optimization. Based on two emission sensitivity simulations, we suggest that the emissions of CH$_4$ from the India region is $51.2 \pm 1.6$ Tg yr$^{-1}$ during the period of 2009-2014.



## 1 Introduction

The Asian Summer Monsoon Anticyclone (ASMA) is a dominant circulation in the Upper Troposphere and Lower Stratosphere (UTLS) in the Northern Hemisphere summer, which extends from Southeast Asia to the Middle East [Webster et al., 1998, Fleitmann et al., 2007]. The Asian monsoons may be classified into a few sub-systems, such as the South Asian Monsoon, which affects the Indian subcontinent and surrounding regions. The monsoon is associated with persistent strong convection over India and the Bay of Bengal, elevated surface heating over the Tibetan Plateau, and orographic uplifting at

the southern/south-western slopes of the Himalayas, which contribute to overall ascension of boundary layer air to the upper troposphere (up to 200 – 100 hPa) [Fu et al., 2006]. The deep convection and associated circulation patterns of the monsoon provides an important pathway for polluted boundary layer air to reach UTLS [Randel and Park, 2006, Randel et al., 2010]. Then atmospheric compounds can be advected over other regions, or further uplifted in the stratosphere [Xiong et al., 2009, Patra et al., 2011, Garny and Randel, 2016]. Due to the influence of deep convection and long-range transport, the chemical

tracers such as $CH_4$, CO, and ozone show sometimes extreme values [Park et al., 2004, 2008].

    The South Asia region, consisting of India, Pakistan, Bangladesh, Nepal, Bhutan, and Sri Lanka, play an import part in the global $CH_4$ budget, as the regional total emission cover 8% of about 500 Tg $CH_4$ global total emissions during the 2000s [Patra et al., 2013]. The recent economic growth of India has led to a significant increase in industrial emissions [Akimoto, 2003, Ohara et al., 2007, Janssens-Maenhout et al., 2019], especial in the Indo-Gangetic Plain (IGP) encompassing northern regions

of India, which is one of the most densely populated region on the globe [Kar et al., 2010]. However, there are little or no long-term measurements of $CH_4$ and the other greenhouse gases to evaluate the inventory emissions [Ganesan et al., 2017, Lin et al., 2018, Chandra et al., 2019].

    Global observations from satellite instruments can complement and extend the information available from the surface *in situ* and aircraft measurements to improve our knowledge of the processes controlling emission and distribution of methane,

to monitor its variability on different scales. The shortwave infrared (SWIR) and thermal infrared (TIR) bands are available for measurements of $CH_4$ from space.

    Observations in the SWIR, such as those from SCIAMACHY on Envisat [Buchwitz et al., 2005, Frankenberg et al., 2011] and TANSO Fourier transform spectrometer (FTS) on GOSAT [Butz et al., 2010, Parker et al., 2011, Yoshida et al., 2013], provide information on column-averaged methane, in cloud-free conditions. SWIR measurements are limited to daytime and

predominantly over land. On the other hand, TIR $CH_4$ observations provide much greater geographical and temporal coverage, and more importantly the measurements of vertical profiles allow a better understanding of the $CH_4$ cycle over a region. The sensitivity of TIR $CH_4$ observations is stronger in the mid to upper troposphere and relatively low near the surface, as its spectral signatures depend on thermal contrast between the atmosphere and surface [Saitoh et al., 2016, de Lange and Landgraf, 2018].

At present moment five nadir-viewing TIR instruments operate on orbit including Atmospheric InfraRed Sounder (AIRS), aboard Aqua, launched in 2002 [Xiong et al., 2008, Zou et al., 2016]; Infrared Atmospheric Sounding Interferometer (IASI),



aboard Metop-A and Metop-B, launched in respectively 2006 and 2012 [Razavi et al., 2009, Crevoisier et al., 2009, 2013, Xiong et al., 2013, Siddans et al., 2017]; TANSO-FTS, aboard GOSAT, launched in 2009 [Kuze et al., 2009, 2012, Yokota et al., 2009, Saitoh et al., 2012, 2016]; Cross-track Infrared Sounder (CrIS), aboard Suomi-NPP, launched in 2011 [Han et al., 2013]; and TANSO-FTS-2, aboard GOSAT-2, launched in 2018 [Matsunaga et al., 2019].

The columnar dry-air mole fractions of methane ($XCH_4$) retrieved over Indian regions from SCIAMACHY shows large spatio-temporal variation closely associated with the distribution of sources like livestock population, wetland, biomass burning, oil and gas production [Kavitha and Nair, 2016]. The seasonal variation of $XCH_4$ is controlled by agricultural activities, mainly rice cultivation as revealed by NDVI analysis [Hayashida et al., 2013]. Along with the heterogeneity in surface emissions variations of $XCH_4$ governed by complex atmospheric transport mechanisms during the southwestern monsoon season in July–September and northeastern monsoon season in October–December as observed by GOSAT. Chandra et al. [2017] have highlighted the difficulty in interpreting the emissions from the surface by columnar $CH_4$ measurements from SWIR spectra, without using an atmospheric chemistry-transport model. At the same time Ricaud et al. [2014] investigated the space-time variations in tropospheric $CH_4$ over the Mediterranean Basin regions using a wide variety of datasets including GOSAT/TANSO-FTS TIR observations.

This study attempts to analyze the vertical distributions of $CH_4$ over the Asian monsoon region. We used $CH_4$ mixing ratios observed from GOSAT/TANSO-FTS TIR (hereafter referred as "GOSAT-TIR") and simulated by the Model for Interdisciplinary Research on Climate (MIROC, version 4.0) [Watanabe et al., 2008] based atmospheric chemistry transport models (ACTM) [Patra et al., 2018] referred to as "MIROC4-ACTM". We aim to understand relative contributions of surface emissions and transport in the formation of $CH_4$ seasonal cycles over different parts of India and the surrounding oceans. The paper is structured as follows. In Section 2, we briefly describe the spaceborne instrument GOSAT-TIR and vertical profiles retrievals of $CH_4$, the MIROC4-ACTM simulation setup, the study domain and data processing. The meteorology and climatology of $CH_4$ inferred from the different data sets over the study domain, variability of $CH_4$ vertical profiles and the impact of the Asian Monsoon Anticyclone to the distribution of the tropospheric $CH_4$ are discussed in Section 3. Major conclusions are given in Section 4.

## 2 Method

### 2.1 GOSAT data

GOSAT is the first satellite dedicated to global observations of greenhouse gases $CO_2$ and $CH_4$ from space [Yokota et al., 2009]. After the launch on 23 January 2009, GOSAT has performed observations on a 666 km sun-synchronous orbit with a 3-day revisit cycle, a 12-day operation cycle, and the local solar time of 13:00 ± 15 min.

The Thermal and Near-infrared Sensor for Carbon Observation Fourier Transform Spectrometer (TANSO-FTS) on board GOSAT detects short-wavelength infrared (SWIR) light reflected from the earth's surface, along with the thermal infrared (TIR) radiation emitted from the ground and atmosphere [Kuze et al., 2009, 2012]. As a result, from these spectral bands,





GOSAT/TANSO-FTS can simultaneously observe $CH_4$ column averaged dry-air mole fractions and $CH_4$ profiles in the same field of view, corresponding to a nadir footprint diameter of 10.5 km. The a priori profiles used in the $CH_4$ retrieval are provided by the National Institute for Environmental Studies (NIES) transport model [Saeki et al., 2013]. Temperature and water vapor profiles necessary for the retrieval are provided by the Japan Meteorological Agency Grid Point Values (JMA-GPV) dataset.

The first retrieval version of the GOSAT-TIR $CH_4$ product (V00.01) and its validation analysis showed the total column values of $CH_4$ ($XCH_4$) based on the GOSAT-TIR $CH_4$ profiles agreed within 0.5% of the aircraft $XCH_4$ values over the tropical ocean [Saitoh et al., 2012]. Holl et al. [2016] did comparisons among $CH_4$ data from ACE-FTS, ground based FTS, and the current released version of GOSAT-TIR ( ) in the Canadian high Arctic, although GOSAT-TIR $CH_4$ measurement information content is too low for a true profile retrieval because of the low thermal contrast and the low signal-to-noise ratio there. Global comparisons with AIRS retrievals reveal good agreement at 300–600 hPa, where both AIRS and GOSAT-TIR $CH_4$ have peak sensitivities [Zou et al., 2016]. Mean mismatch in $CH_4$ (GOSAT–AIRS) were 10.3−31.8 and -16.2±25.7 ppbv for the levels of 300 and 600 hPa, respectively. Comparison of the $XCH_4$ shows that GOSAT-TIR agrees with AIRS to within 1% in the mid-latitude regions of the Southern Hemisphere and in the tropics. However, disagreement increases to ∼1-2% in the mid to high latitudes [Zou et al., 2016]. Olsen et al. [2017] performed global comparisons of GOSAT-TIR vertical profiles to the Canadian Space Agency's Atmospheric Chemistry Experiment FTS (ACE-FTS) on SCISAT (version 3.5) and the European Space Agency's Michelson Interferometer for Passive Atmospheric Sounding (MIPAS) on Envisat, as well as 16 ground stations with the Network for the Detection of Atmospheric Composition Change (NDACC).

In the overlapping altitude ranges of the three satellite data products there is a small, but consistent, positive bias of around 20 ppbv, or 1% in GOSAT-TIR $CH_4$ data. In the upper troposphere, good agreement between TANSO-FTS and NDACC was found, without a bias. In a more recent comparison, the average bias in $CH_4$ profile retrieved from GOSAT-TIR spectra with a spectral correction scheme is less than 2% over the full altitude range, when compared with data from the Monitoring Atmospheric Composition and Climate (MACC) scaled to the total column measurements of the Total Carbon Column Observing Network (TCCON) [de Lange and Landgraf, 2018].

This study uses the GOSAT-TIR $CH_4$ product, which is released for the period from April 23, 2009, through May 24, 2014. The number of vertical grid layers of the GOSAT-TIR $CH_4$ product is 22 from the surface to 0.1 hPa.

### 2.2 MIROC4-ACTM simulations

The measurements are compared to results of simulations by the MIROC4-ACTM chemical tracer simulation [Watanabe et al., 2008, Patra et al., 2018]. The MIROC4-ACTM runs at a horizontal resolution of T42 spectral truncations (≈2.8 ×2.8) with 67 sigma-pressure vertical levels. The MIROC4-ACTM simulated horizontal winds (U and V) and temperature (T) are nudged to the Japan Meteorological Agency reanalysis fields (JRA-55) at all the vertical levels [Kobayashi et al., 2015]. The model uses an optimal OH field based on a scaled version of the seasonally varying OH field [Patra et al., 2014].



Two simulations were performed using combinations of inverted fluxes based on the following *a priori* emission scenarios prepared on a monthly basis by combining the emissions from all anthropogenic and natural sectors, and by subtracting the surface sinks due to bacterial consumption in the soil [Chandra et al., 2020]:

   1. $\text{Flux}_{\text{Cao}}$: EDGAR + GFED + other + VISIT wetland (Cao scheme [Cao et al., 1996]).

   2. $\text{Flux}_{\text{WH}}$: EDGAR + GFED + other + VISIT wetland (WH scheme [Walter et al.,2001]).

Here, EDGAR (the Emission Database for Global Atmospheric Research, version 4.3.2 inventory [Janssens-Maenhout et al., 2019]) provides the anthropogenic emissions for individual sector at the spatial resolution of $0.1°\times0.1°$on an annual basis for the period 1970-2012. The growth rate for 2011-2012 has been used to interpolate the inventory emissions for 2013-2017. GFED (Global Fire Database version 4s [van der Werf et al., 2017]) is monthly emissions for biomass burning. The other emissions (including ocean, termites, mud volcano etc.) are taken from TransCom-CH4 inter-comparison experiment [Patra et

al., 2011]. VISIT (Vegetation Integrated Simulator of Trace gases [Ito, 2019, Ito et al., 2019]) is a process-based model of the terrestrial biogeochemical cycle, which is used to estimate the wetland and rice emissions on monthly basis using two different schemes developed by Cao et al. [1996] and Walter et al. [2001], referred by "$\text{ACTM}_{\text{Cao}}$" and "$\text{ACTM}_{\text{WH}}$" scheme, respectively.

### 2.3 Data processing

The MIROC4-ACTM data were collocated to the GOSAT-TIR observation points. The criteria for the collocation are the
nearest model grid cell in space, and the nearest hour in time. For vertical profile comparison, the MIROC4-ACTM data were interpolated on the retrieval pressure levels of the GOSAT-TIR product, i.e. from 67 to 22 levels.

### 2.4 Averaging kernels and the retrieval sensitivity

The averaging kernels (AK) are defined to provide a simple characterization of the relationship between the retrieval and the true state. The retrieval sensitivity can be obtained from the sum of the columns of the averaging kernel matrix, which is
also referred to as "the area of the averaging kernel" [Rodgers, 2000].

Along with "raw" model simulation results ($\text{ACTM}_{\text{Cao,WH}}$) we analysed ($\text{ACTM}_{\text{Cao,WH}}^{\text{AK}}$) profiles convoluted with retrieval *a priori* and the GOSAT-TIR CH4 averaging kernel matrix using the following vector equation [Rodgers, 2000, Saitoh et al., 2012]:

$$\mathbf{X}_{\text{ACTM}_{\text{Cao,WH}}^{\text{AK}}} = \mathbf{X}_{a\ priori} + \mathbf{A}(\mathbf{X}_{\text{ACTM}_{\text{Cao,WH}}} - \mathbf{X}_{a\ priori}) \qquad (1)$$

Here, $\mathbf{A}$ is an averaging kernel matrix, $\mathbf{X}_{apriori}$ represent a vector of *a priori* vertical profile, $\mathbf{X}_{\text{ACTM}_{\text{Cao,WH}}}$ and $\mathbf{X}_{\text{ACTM}_{\text{Cao,WH}}^{\text{AK}}}$ are vectors of "raw" and convoluted model simulated profiles, respectively.

### 2.5 Study domain

This work follows the setup described by [Chandra et al., 2017] and uses 10 regions (Fig. 1) which are characterized by different CH4 emission and meteorological conditions. The Indian landmass was partitioned into eight sub-regions surrounding



by two oceanic regions. From Southwest to Northeast there are: the Arabian Sea (AS), Southern India (SI), the Bay of Bengal (BB), Western India (WI), Central India (CI), Eastern India (EI), Arid India (AI), Western IGP (WIGP), Eastern IGP (EIGP), and Northeast India (NEI).

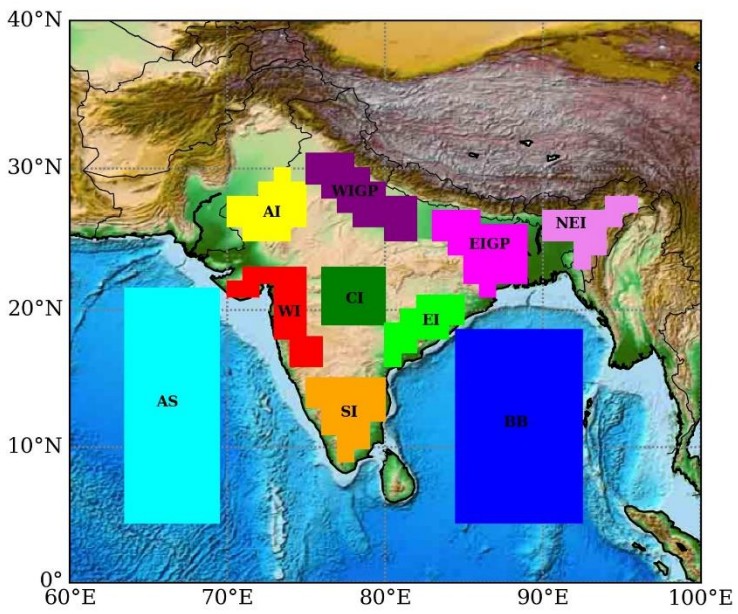

**Figure 1: The map of the regional divisions for the analysis, as defined by [Chandra et al., 2017]. The Indian regions from**
**Southwest to Northeast are following: the Arabian Sea (AS), Southern India (SI), the Bay of Bengal (BB), Western India (WI), Central India (CI), Eastern India (EI), Arid India (AI), Western IGP (WIGP), Eastern IGP (EIGP), and Northeast India (NEI). Here and after all countries' borders are plotted using maps build in the matplotlib basemap toolkit (v. 1.2.0) in Python.**

## 3 Results and discussion

### 3.1 Atmospheric conditions controlling the spatial distribution of methane

Key-components of the climatology in the Indian Ocean and the surrounding areas are the annual migration of the Intertropical Convergence Zone (ITCZ) and seasonal development of the monsoon winds [Findlater, 1969, Webster et al., 1998, Fleitmann et al., 2007]. In boreal spring the ITCZ migrates northward across the Indian Ocean and reaches its northernmost position at approximately 35 N during summer. A strong pressure gradient between the low-pressure zone over the Tibetan Plateau and a high-pressure zone over the Southern Indian Ocean generates a strong near surface monsoonal airflow
from July to September (Fig. 2c1-c3).

    In autumn the ITCZ then retreats southward and reaches its southernmost position at approximately 25 S in January. The reversed pressure gradient during the winter months generates the moderate and dry northeast monsoon (Fig. 2d1-d3). Using Moderate Resolution Imaging Spectroradiometer (MODIS) onboard Terra and Aqua satellites Devasthale and Fueglistaler [2010] showed that a significant fraction of high opaque clouds reaches and penetrates the tropical tropopause layer during



**Figure 2:** The wind speed vector fields for the JFM (panels a1-a3), AMJ (b1-b3), JAS (c1-c3), and OND (d1-d3) periods of 2010 from the MIROC4-ACTM dataset. Please note different vector scale for the levels of 800 (panels a1-d1), 500 (a2-d2), and 200 hPa (a3-d3) respectively. At the background: left panels (a1-d1) show monthly surface pressure (0.9950 sigma level) from the National Centers for Environmental Prediction (NCEP) reanalysis averaged for 1981-2010 (http://www.esrl.noaa.gov/psd/data/ gridded/data.ncep.reanalysis.derived.html), central panels (a2-d2) show monthly long term mean interpolated outgoing longwave radiation (OLR) from National Oceanic and Atmospheric Administration (NOAA) averaged for 1981-2010 (https://www.esrl. noaa.gov/psd/data/gridded/data.interp_OLR.html), the right panels (a3-d3) show seasonally averaged (for 2009-2014) daily mean cloud top pressure (hPa) from the level-3 MODIS atmosphere daily global product (v6.1) downloaded from the Giovanni online data system [Acker and Leptoukh, 2007].

active monsoon conditions Fig. 2c3. The overall frequency of convective clouds (reaching at least 200 hPa) is higher in July and August. Most of the deep convection occurs over the Bay of Bengal and central northeast India Bergman et al. [2013].



They suggested that very deep convection over the Tibetan plateau is comparatively weak, and may play only a secondary role in troposphere-to-stratosphere transport.

### 3.2 GOSAT-TIR CH$_4$ profile properties

The observation by GOSAT-TIR band enables us to analyse the vertical structure of atmospheric CH$_4$. This band has relatively high spectral resolution of ~0.2 cm$^{-1}$ and provides CH$_4$ vertical profiles in 22 layers. The degrees of freedom of signal for CH$_4$ observation by GOSAT-TIR band (V1 algorithm version) is around 1 over low-latitude part of India. Figure 3 suggests that the GOSAT-TIR spectra are sensitive to the CH4 concentrations in the height range of 900 hPa to 30 hPa. The spectra sensitivity does not change significantly between the different part of our analysis region, as seen from Fig. 3.


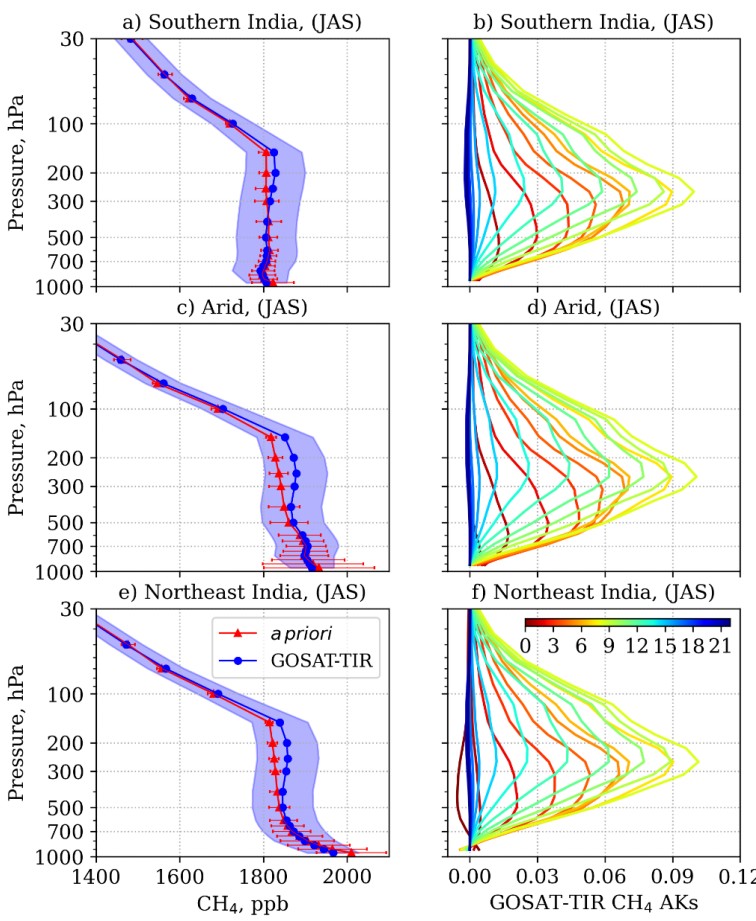

**Figure 3. Seasonal mean (July-September 2011) over regions of Southern India (upper panel), Arid India (middle panel), and Northeast India (bottom panel). The left columns show the GOSAT-TIR CH$_4$ *a priori* with 1-$\sigma$ STD uncertainty (red line with error bars) and GOSAT-TIR CH$_4$ profile with the retrieval error (blue line with shaded area); the right columns depict the averaging kernels of GOSAT-TIR CH4 retrievals averaged over time. There are 22 lines for GOSAT-TIR retrievals, corresponding to the retrieval layers used in each of them.**



Through analysis of the AK profiles of the GOSAT-TIR V1 $CH_4$ products, Zou et al. [AMT, 2016] show the sensitivity of GOSAT-TIR measurements gradually increases from the surface up and reaches a maximum at the levels of 300–600 hPa in the high latitudes and 200–600 hPa in the tropics. While below 800 hPa the sensitivity is small reflecting the major limitation of TIR in measuring the change of $CH_4$ in the lower troposphere. Figure 3 shows typical GOSAT-TIR $CH_4$ profile with retrieval errors and *a priori* profile (left panels) and its corresponding AK profiles (right panels) for the Southern, Arid, and Northeast regions during the monsoon season. At the pressure levels of 500–150 hP where GOSAT-TIR measurements have sensitivity judging from their AK profiles, there are some differences between the retrieved and a priori CH4 profiles and they are beyond the retrieval errors, which means they should be significant differences.

### 3.3 CH₄ over India observed by GOSAT-TIR and simulated by MIROC4-ACTM

In this section we analyzed $CH_4$ distributions from GOSAT-TIR and MIROC4-ACTM at the levels of the constant pressure of 800, 500, and 200 hPa, which represents the top of the boundary layer, the free troposphere, and the upper troposphere parts of the atmosphere, respectively. Datasets were resampled on the grid with a resolution of 3.0 ×3.0 and interpolated along the vertical coordinate. $CH_4$ concentrations were average in time for three periods: pre-monsoon April-June (AMJ), monsoon July-September (JAS), and post-monsoon October-December (OND). The winter season (JFM), as no strong difference found in comparison with AMJ.

Due to a lack of GOSAT-TIR $CH_4$ data in cloudy scenes and the influence of the complex orography of the studied area, the number of points used for averaging in each grid cell varies with height over land (Fig. 4d1-d3). This is especially noticeable for the northern regions of India, since a significant part of Tibet and the Himalayas are above the level of 800 hPa (Fig. 4d1). Northern India also has large sources of $CH_4$ with different types of emission. These two factors cause large standard deviations (STD) in $CH_4$ (Fig. 4e1-43). For South India and the marine regions, the STD values are much lower compared to those over the land.

In the middle and upper troposphere, the perturbations from the heterogeneity of the emissions are smoothed out, the density of observation points increases, therefore, the averaging errors decrease. At a height of 200 hPa, the average STD for GOSAT-TIR is approximately 25 ppb.

The density of observation points decreases with the onset of the monsoon season (Fig. 5d1-d3), however, it remains sufficient to detect significant changes in $CH_4$ concentrations even considering the relatively large STD values there. A significant increase in concentration values is noticeable primarily in the middle and upper parts of the atmosphere (Fig. 5a1-a3), which is due to the repeatedly confirmed effect of convective transport from surface sources upward. After reaching a level near the tropopause, the increased concentrations are distributed by three jets: the lateral (the cross-equatorial circulation) and transverse (flows between the arid regions of north Africa and the Near East and south Asia) monsoons, and the Walker Circulation is extended across the Pacific Ocean [Webster et al., 1998]. The $CH_4$ concentration in the eastern jets is higher, since it is formed over more northern areas with larger emission. The influence of the third component (the cross-equatorial circulation) is more noticeable in the post-monsoon period (Fig. S2).





**Figure 4: Latitude-longitude distributions of CH₄ at the levels of 800, 500, and 200hPa (the left, middle, and right panels respectively) observed by GOSAT-TIR for the season AMJ 2011. The first upper panels (a1-a3) show GOSAT-TIR CH₄, the second upper panels (b1-b3) show GOSAT-TIR *a priori* CH₄, the third upper panels (c1-c3) the GOSAT-TIR observation points numbers, the fourth upper panels (d1-d3) show GOSAT-TIR CH₄ standard deviation, respectively.**

GOSAT-TIR $CH_4$ retrievals are constrained to the a priori $CH_4$ data (panels b1-b3 of Fig. 3–4) especially in lower pressure levels due to the relatively low signal-to-noise ratio of the TIR spectra at $CH_4$ absorption bands [Saitoh et al., 2012, Zou et al., 2016]. Nevertheless, the GOSAT-TIR $CH_4$ product shows vivid differences in $CH_4$ from the *a priori* values even in the lower





part of the atmosphere, where sensitivity is weak (panels c1-c3 of Fig. 3–4). This implies an additional signal of CH$_4$

concentration could be captured by the GOSAT-TIR measurements.



**Figure 5: Same as Fig. 3, but for JAS 2011.**

As explained in Section 2.2, MIROC4-ACTM simulations were performed with two flux combinations reflecting different

approaches for estimation of the wetland CH$_4$ emission. In general, the WH scheme fluxes are about 5-10% larger the Cao,





excepting the WIGP, EIGP, and NEI regions of India and Bangladesh where the maximum difference reaches 20-40% (Fig. 6). Besides, there are small hot spots in Southeast Asia (e.g. Mekong River Delta).

Since in the pre- and post-monsoon seasons (AMJ and OND) the excess concentration due to additional emission is locked in the boundary layer (as seen from MOPITT CO) [Kar et al., 2010], we can detect only a slight increase in concentration at

the levels selected for the analysis. $CH_4$ simulated using both emission schemes are consistent with the GOSAT-TIR retrieval with averaged mismatch within $\pm2\%$, the heterogeneity of which is apparently caused by transport regimes (see Fig. 7 for AMJ). By analogy to the $CH_4$ distribution from GOSAT-TIR the increased scatter found in modeled $CH_4$ over IGP, wherein the enhanced values extend up to the level of 200 hPa (see supplement Fig. S4).

During the monsoon, the difference between emission scenarios becomes significant, as additional $CH_4$ mass is carried to

the middle and upper atmosphere (Fig. 8). The larger mismatch in comparison with GOSAT-TIR (Fig. 8b1-b3) emphasizes the redundancy of CH4 emission of the WH scheme.

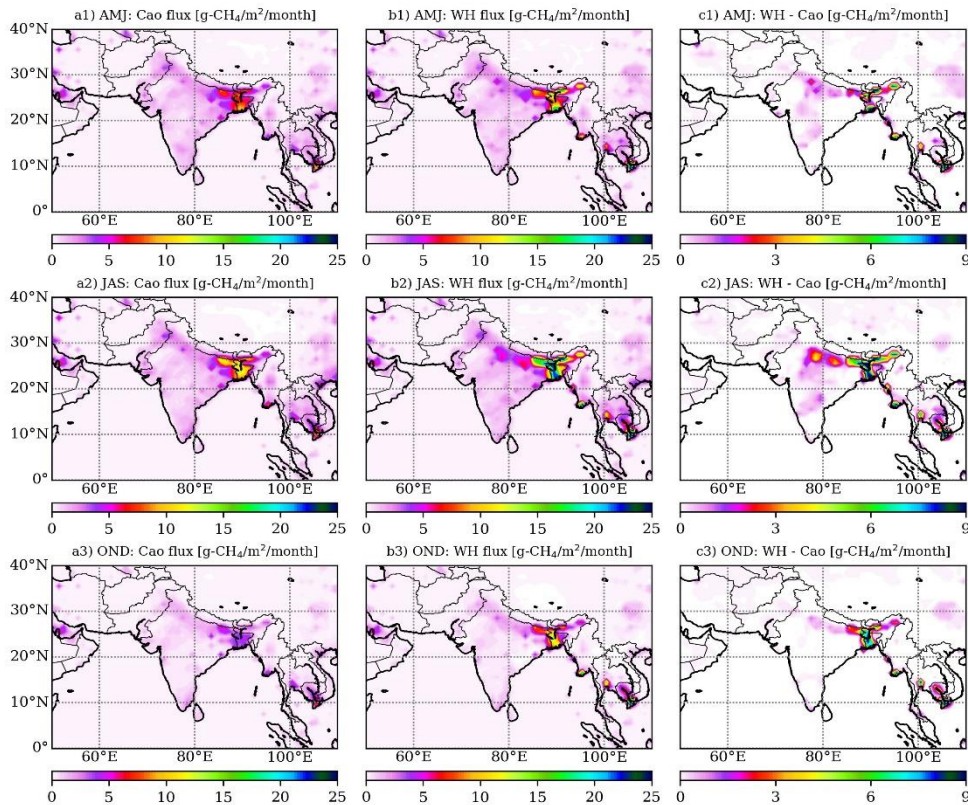

**Figure 6: The surface $CH_4$ fluxes (g-$CH_4$/m²/month) used for MIROC4-ACTM simulation: a1a3) from Cao scheme, b1-b3) from WH scheme, and c1-c3) difference between schemes. Panels a1-c1, a2-c2, and a3-c3 are for AMJ, JAS, and OND respectively.**


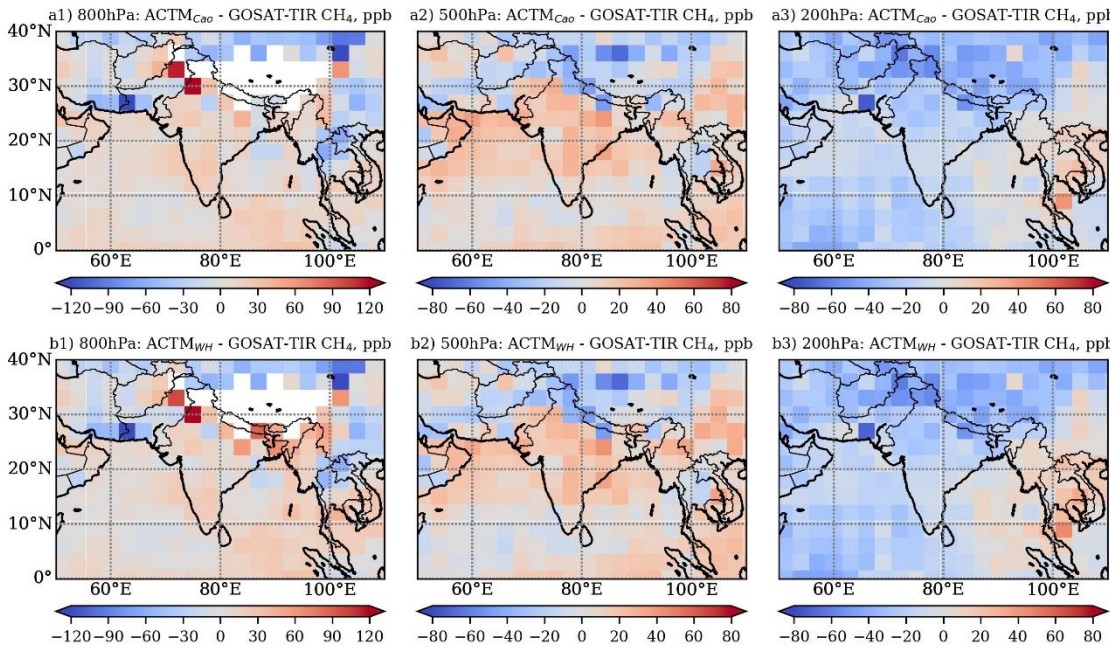


**Figure 7: Latitude-longitude distributions of CH₄ simulated by MIROC4-ACTM at the levels of 800, 500, and 200 hPa (the left, middle, and right panels respectively) for AMJ 2011. The first (a1-a3) and second (b1-b3) upper panels show the difference in CH₄ between GOSAT-TIR and ACTM_Cao and ACTM_WH. The averaging kernel was not implemented.**

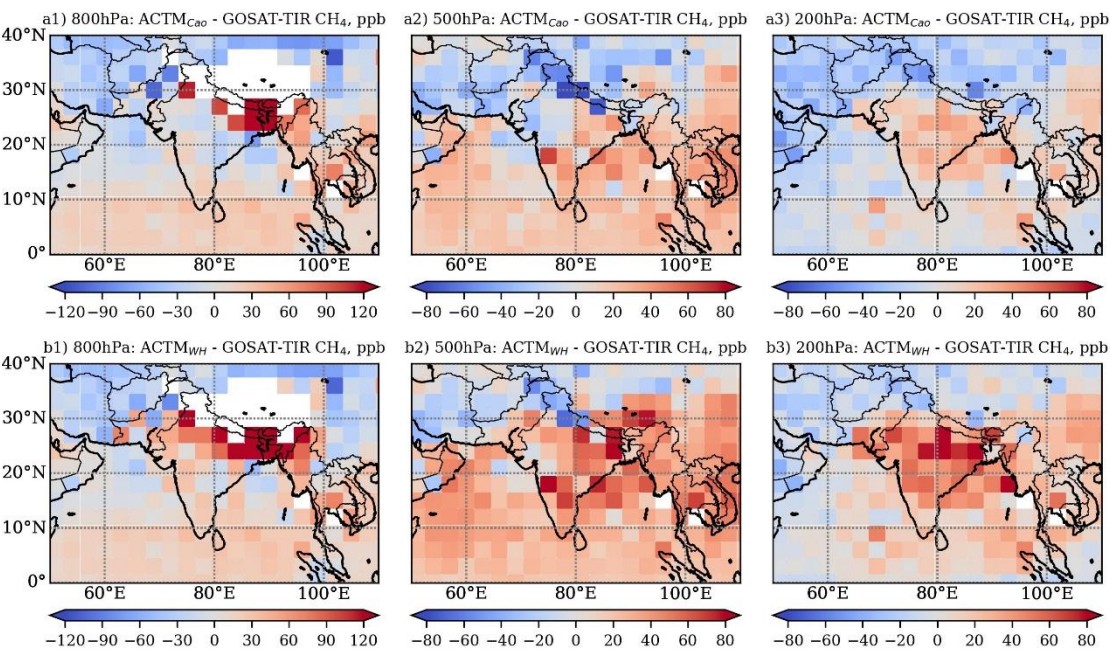

**Figure 8: Same as Fig. 6, but for JAS 2011.**





### 3.4 CH₄ vertical profiles

This section aiming to investigate the $CH_4$ tropospheric profile time and space variations above the Indian regions and to attribute the altitude $CH_4$ variability to the regional emission strength and different synoptic and global scale depending on the season. Figure 9 depicts seasonal mean $CH_4$ vertical profiles observed by GOSAT-TIR and simulated by the model for pre-monsoon (April-June) and monsoon (July-September) of 2011. The variation of GOSAT-TIR sensitivity are taking into account by the implementation of the averaging kernel the modelled data sets ($ACTM^{AK}_{Cao}$ and $ACTM^{AK}_{WH}$).


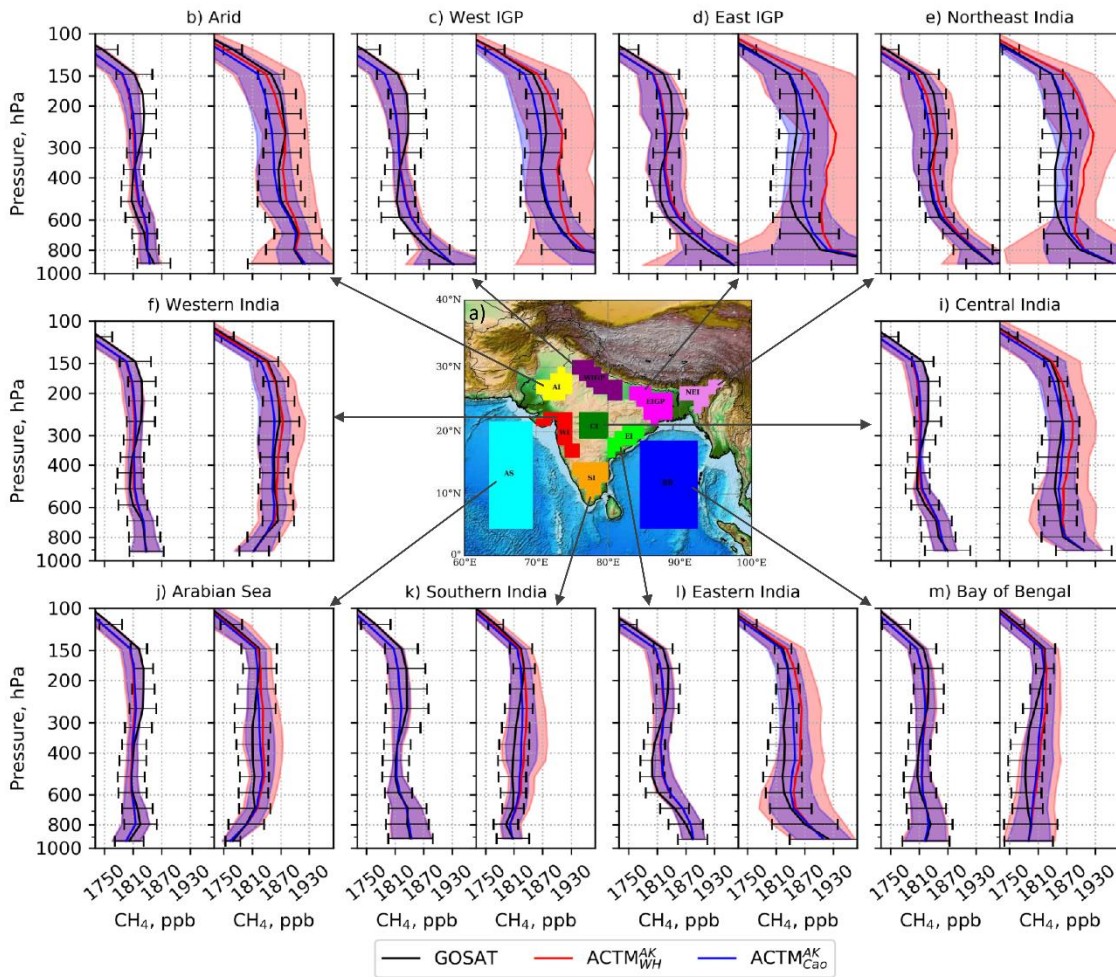

**Figure 9: Seasonal mean CH₄ vertical profiles for pre-monsoon (April-June) and monsoon (July-September) of 2011 are shown in the left and right part of each panels (b-m) for the different regions as depicted in the central map (a). Black line with error bars shows the GOSAT-TIR data with 1-σ STD uncertainty. Blue and red lines with shaded areas correspond to the ACTM^{AK}_{Cao} and ACTM^{AK}_{WH} data with 1-σ STD uncertainty, respectively.**






By using the a priori and retrieved $CH_4$ profiles with retrieval error and AK profile (Fig. 3), we found that differences

between *a priori* and retrieved $CH_4$ profiles are larger than its retrieval error, so the differences are valid to be discussed. The

variabilities shown in Figure 9 are larger than GOSAT-TIR retrieval errors, so GOSAT-TIR and model show good agreements

within both errors (natural variabilities and retrieval random errors).

The vertical $CH_4$ profiles have a characteristic curved shape with double peak. The first peak near the surface is associated

with emissions from local sources, the second one at the level of 150-200 hPa is caused by the vertical updraft [Belikov et al.,

2013, Saito et al., 2013]. Reflecting the increase of $CH_4$ surface fluxes intensity (Fig. 6), the vertical gradient between the near-

surface and upper troposphere levels increases in the direction from South-West (marine regions (Fig. 9a,c) have slightly lower

concentrations in the boundary layer since the sea is a weak source) to the North-Eastern (where EIGP, WIGP, and Northeast

Indian stand out in significant sources due to various natural and anthropogenic sources (Fig. 9h,i,j)).

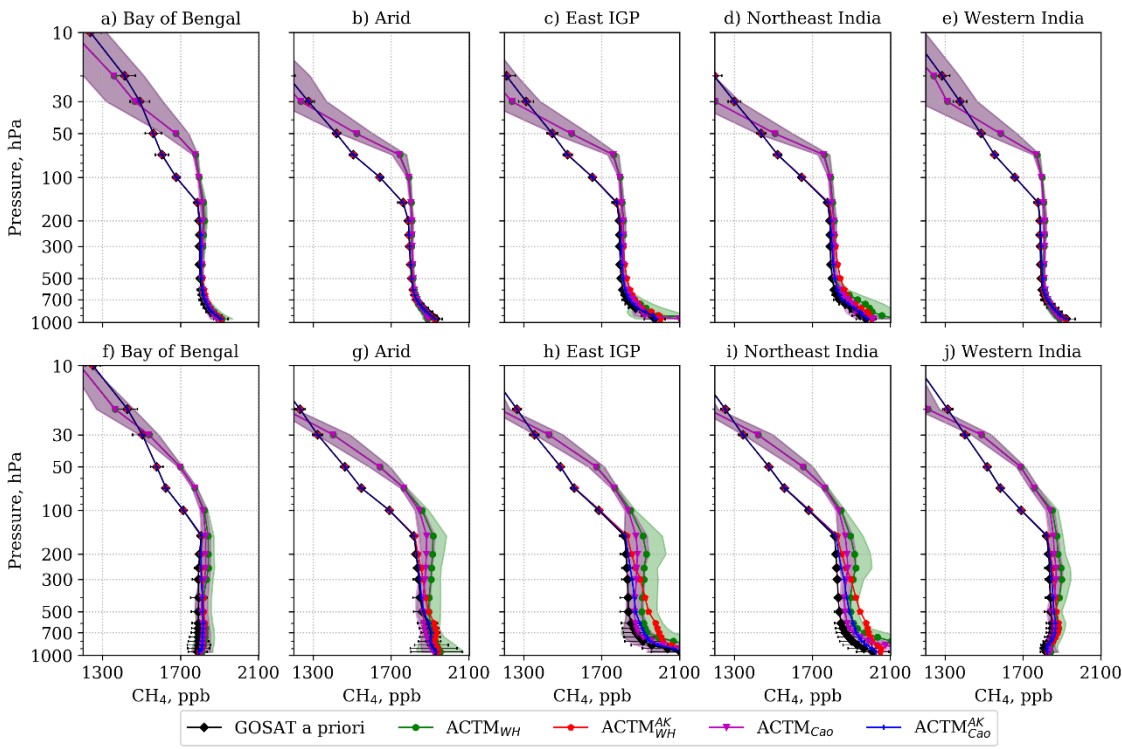


**Figure 10: Seasonal mean $CH_4$ vertical profiles (a-e) for winter (January-March) and (f-j) monsoon (July-September) periods of 2011 are shown for the selected regions. Black line shows the GOSAT-TIR a priori data. Green and magenta lines with shaded areas correspond to the $ACTM_{WH}$ and $ACTM_{Cao}$ data with 1-$\sigma$ STD uncertainty, respectively. Red and blue lines represent the $CH_4$ from $ACTM^{AK}_{WH}$ and $ACTM^{AK}_{Cao}$ smoothed with the averaging kernel implementation.**


Monsoons cause a powerful perturbation of concentration along the entire vertical profile up to the level of the tropopause.

Two southern regions (the Arabian Sea and Southern India; Fig. 9a,h) are located near the entry point of Somali Jet -



atmospheric masses with a low $CH_4$ coming from the Indian Ocean [Findlater, 1969]. These regions do not have significant sources of $CH_4$, and therefore, concentration in the vertical profiles increase with height due to transport from other regions.

The third southern region (Bay of Bengal; Fig. 9c) has similar properties, but at the same time, it is under the influence of transport from neighboring regions (i.e. East India, EIGP), as evidenced by a large spread near the surface.

The use of AK is taking into account the relatively low vertical resolution of satellite measurements and the change in the sensitivity of the retrieval by smoothing along the a priori profile and reduces the spread at the levels where the sensitivity of satellite sensors is weak. Convolution of modelled profiles with GOSAT-TIR $CH_4$ averaging kernels (Eq. 1) smooths the model

profiles to fit the GOSAT-TIR vertical resolution and reduce their mismatch. Fig. 10 shows GOSAT-TIR AK has significant smoothing, approaching the MIROC4-ACTM model profiles to *a priori* so much that the difference between the calculations for the Cao and WH emission scenarios becomes barely distinguishable. This is especially vivid above the level of 150 hPa, where the sensitivity of GOSAT-TIR there drops sharply and the satellite retrievals and the AK convolved model profiles strongly follow the *a priori* profiles.

The choice of an a priori profile (usually provided by model calculations) is an important point in retrieval problems. The TransCom-CH4 experiment [Patra et al., 2011] showed a significant scatter between the participated models, including the NIES model later selected for calculating GOSAT-TIR a priori profile. In our study, a significant difference in the methane profile gradient, its seasonal variability (winter and summer) between a priori and the MIROC4-ACTM model was revealed in UTLS zone (levels of 150-20 hPa). Apparently, the difference in modelling the tropopause region and the tracer transport

into the lower stratosphere is a key factor. Here should be noted that MIROC4-ACTM uses a more modern reanalysis to calculate the meteorological parameter, and the vertical resolution (67 sigma-pressure levels) is quite higher than that of the NIES (47 sigma levels). Even more important, the stratospheric part of the NIES model was adjusted to observed age of air for $CO_2$ and long-term satellite observations from HALOE for $CH_4$ [Saeki et al. 2013]. This emphasizes the uncertainty in modelling transport processes near the tropopause derived by different methods.

From the moment GOSAT was launched, the calculation of a prior profiles is carried out according to the same scheme. This is important for the long-term consistency of the GOSAT-TIR $CH_4$ product but does not take into account the significant improvements (for example, updated OH fields, reanalysis, convective parameterization) implemented for MIROC4-ACTM. This emphasizes the need to use custom *a priori* profiles in retrieval, which requires greater transparency of technical information from satellite projects. This problem is less noticeable, but no less relevant for satellite $CH_4$ receivers operating in

the SWIR band aiming to obtain the total column $CH_4$.

### 3.5 $CH_4$ time-altitude variation

The monsoon anticyclone shows substantial intra-seasonal oscillations, which are connected to variable forcing from transient deep convection over the Indian subcontinent and the Bay of Bengal. This variability is typically associated with active/break cycles of the monsoon with timescales of ∼10–20 days. Significant correlations exist between outgoing longwave

radiation (OLR; Fig. 2a2-d2) and circulation within the monsoon region, such that the entire balanced anticyclone varies in





concert with convective heating: enhanced convection leads to warmer tropospheric temperatures, stronger anticyclonic circulation, and colder lower stratospheric (and tropopause) temperatures [Randel and Park, 2006]. This causes a significant heterogeneity of the flux transported upward and $CH_4$ concentration in the upper layers during ASMA (Fig. 11).

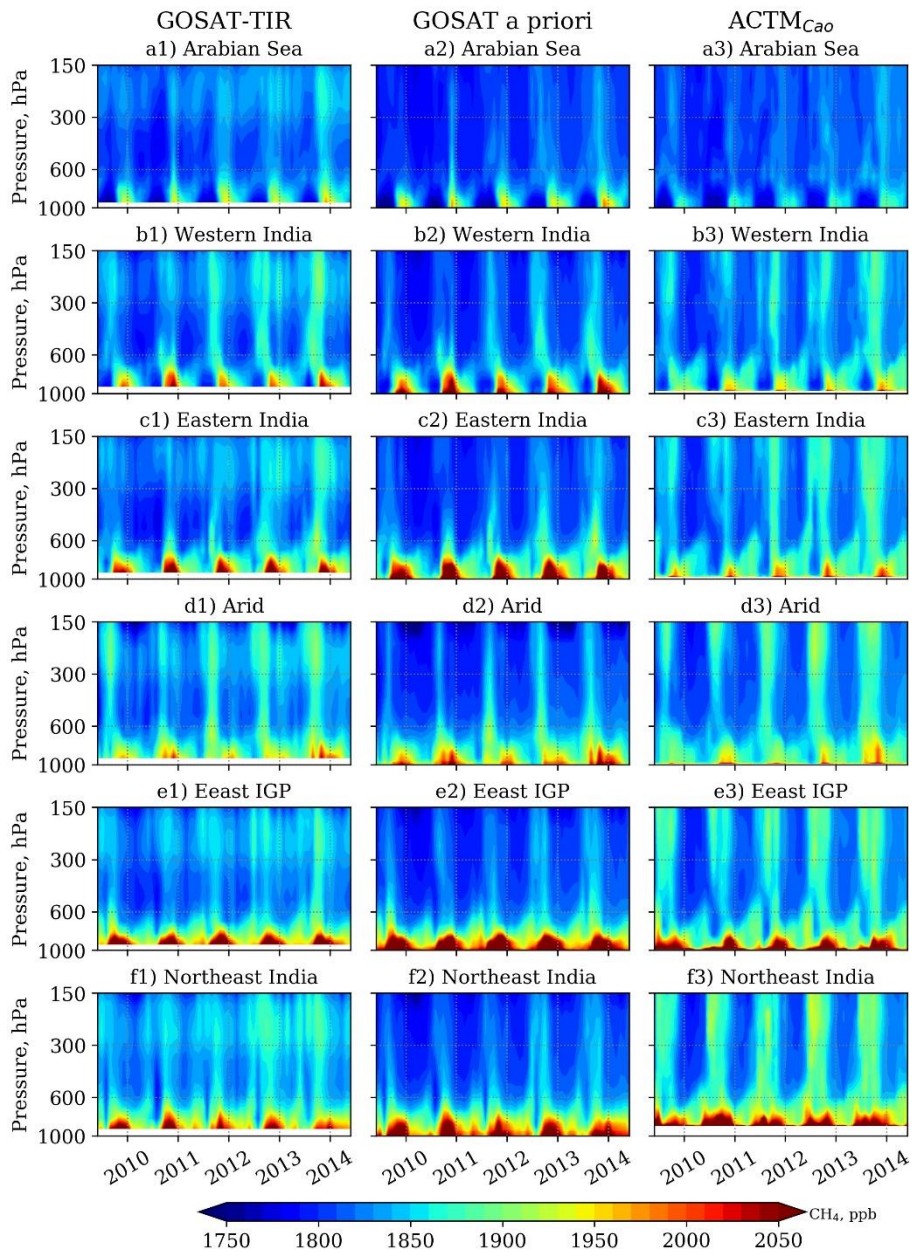

**Figure 11: Time-altitude cross-section of CH4 form GOSAT-TIR retrieval, GOSAT-TIR *a priori* and ACTM$_{Cao}$ (the left, middle, and right panels respectively) for considered regions. Note that the profiles are shown for the tropospheric altitudes as the GOSAT-TIR retrieval system is not sensitive to the stratospheric altitudes (See Fig. 9 and the associated text).**





The IGP region experiences intense agricultural activity, and use of traditional biofuels. In the winter months the IGP is often enveloped by thick fog and haze (Gautam et al., 2007). The prevailing winds at low altitudes (surface to ~850 hPa) are

northerly to northwesterly with low wind speeds (<5 m/s) and the eastern parts of the IGP are impacted by a localized area of strong subsidence in winter [Dey and Di Girolamo, 2010]. These conditions tend to trap the pollution at low altitudes [Kar et al., 2010].

**3.5 Seasonal variation of CH₄**

The Prophet time-series analysis and forecasting model [Taylor and Letham, 2018] was implemented to derive multi-year

(2009-2014) seasonal variation of CH₄ from GOSAT-TIR and MIROC4-ACTM. The model Prophet performs smoothing of time-series data based on a generalized additive model with three main components: trend, seasonality, and holidays. Compared to traditional exponential smoothing, Prophet can easily handle temporal patterns with multiple periods and has no requirements regarding the regularity of measurement spacing. The model has a robust performance in the presence of missing data and trend shifts and typically handles outliers well while working with time-series that have several seasons of historical

data with strong seasonal patterns. The Prophet allows the use of all data points for the study period, thereby increasing accuracy and reducing sensitivity to random outliers [Belikov et al. 2019]. Though the GOSAT-TIR and MIROC4-ACTM mismatch in trend is almost negligible, the difference in the simulation of the amplitude and phase of the seasonal variation can be significant (Fig. 12).

Seasonal changes are controlled primarily by meteorological parameters, so the most noticeable effect is determined by the

summer monsoon. During this season enhanced transport redistributes CH₄ along all layers of the troposphere. The minimum CH₄ seasonal variation is found in the lower troposphere (800 hPa), while the maximum occurs in the upper part (Fig. 12e-f).

The amplitude of seasonal changes is determined by the net amount of the sources; therefore, it increases from south to north from marine regions to the most densely populated areas. The figure 12 shows that a significant difference between the Cao and WH fluxes is evident in the three northern regions (WIGP, EIGP and NEI). During summer their differences can

reach almost 50%. This inequality determines the difference between seasonal variability not only for these regions, but also for the nearest neighbours. Especially noticeable for AI and WI, where intrinsic fluxes are much small.

With the onset of autumn, the deep convective transport is suppressed, therefore under the influence of the Hadley cell circulation the slow outflow of air masses is started in the opposite south-west direction. This moment is characterized by the peak of concentration at 800 hPa, which slowly moves from the northern regions (over EIGP in October) to the southern (over

Arabian Sea in the late November).

India occupies a large region of South Asia, where a fewer observations limit the chance to reduce the uncertainty in the greenhouse gases (including methane) flux. Used in this work the Cao and WH flux combinations for the South Asia region for the period 2009-2014 account $65.7 \pm 2.1$ and $82.4 \pm 2.8$ Tg yr$^{-1}$ respectively. In order to identify which emission scenario is more realistic, we compared the monthly mean methane concentrations averaged over the region's surface area Fig. 13. For



all considered levels, ACTM$_{WH}$ is superior to ACTM$_{Cao}$. However, a comparison with GOSAT-TIR may lead to a slightly different results depending on the level selected for comparison.

**Figure 12:** Seasonal variation of CH$_4$ (right y-axis) derived for levels of 800 (red lines) and 200 hPa (blue lines) over considered regions from GOSAT-TIR (solid line), ACTM$_{Cao}$ (dashed line), and ACTM$_{WH}$ (dotted line), respectively. At the background bar plots 380 represent Cao and WH CH$_4$ fluxes (left y-axis). Please note the different scale of y-axes (left) for fluxes.





Again three levels (800, 500, 200 hPa) were considered. At the top of the boundary layer (Fig. 13c) GOSAT-TIR shows significant inter-seasonal variability, which can be greatly influenced due to the large spread (large STD values) of individual samplings (Fig. 4–5 panels d1-d3)). Another important factor is the GOSAT-TIR retrieval *a priori* profile derived from NIES TM with the coarse vertical grid and simplified scheme for modeling of the boundary layer height, which shows strong diurnal

and seasonal variations [Kavitha et al., 2018]. In UTLS significant seasonal fluctuations also occur (Fig. 13a).

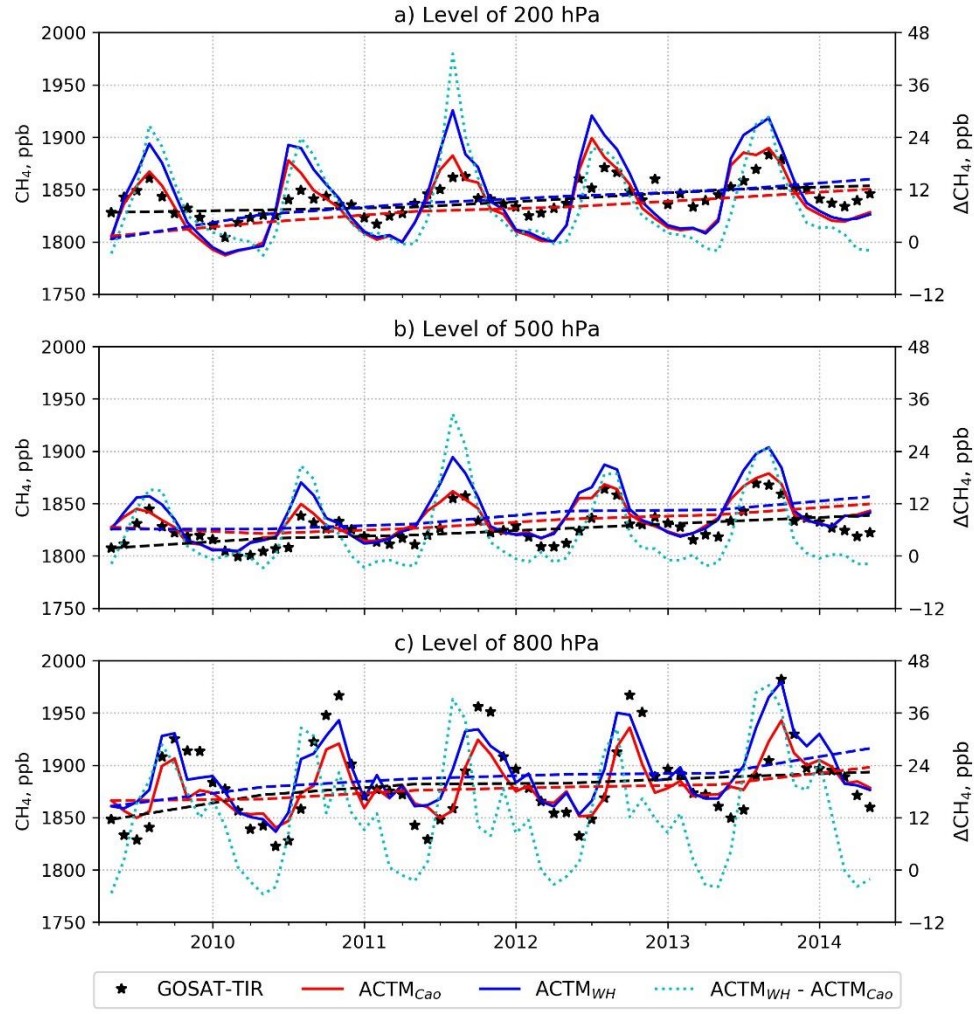

**Figure 13: Time series of CH₄ averaged over the area of South Asia for levels of a) 200, b) 500 and c) 800 hPa respectively. Symbols state the GOSAT-TIR observations, red and blue lines are for ACTM$_{Cao}$, and ACTM$_{WH}$, respectively. Solid and dashed**
**lines are for monthly and yearly averaged concentrations (left y-axis), dotted line shows the difference between the model simulations (right y-axis), respectively. The averaging kernel was not implemented.**



The strong summer peak in the MIROC4-ACTM $CH_4$ is associated with excessive vertical transport, which, apparently was not completely resolved upon the transition to the new (MIROC-4.0) meteorology. Moreover, it remains unclear what

causes significant drops in concentration in the winter period. In the middle troposphere (Fig. 13b) a good consistency in phase is found and the $ACTM_{Cao}$ to $ACTM_{WH}$ concentration mismatch is strongly associated with the flux difference. Relying on this comparison results, we can suggest the Cao flux combination as more reliable emission estimation. This confirm the assessment made by Patra et al. [2016], indicating that the EDGAR inventory (version 4.2FT2010) overestimated the South Asia regional emission by 10-15 Tg yr$^{-1}$. A significant part of the extra fluxes is concentrated in a few relatively small regions

in the Northen India (fig. 6). However, our best estimate emission of 51.2 ± 1.6 Tg yr$^{-1}$ over the India is much greater than those estimated by [Ganesan et al., 2017], combined *in situ* data of different time coverage and SWIR $CH_4$ retrievals in trajectory-based modelling framework.

## 4 Conclusions

Vertical profile observations of $CH_4$ from GOSAT-TIR at 22 pressure layers and simulations by MIROC4-ACTM, sampled

at the location and time of the satellite overpass, were analyzed over India and surrounding oceanic regions for the period 2009-2014. The area of our analysis is subdivided in to several land and ocean regions. The main highlights of the present study are summarized below:

1.   GOSAT-TIR observations provide data coverage and density suitable to study detailed horizontal features of $CH_4$ at the top of the atmospheric boundary layer (excepting high mountain regions), free troposphere, and upper troposphere.

While [Chandra et al., 2017] mainly used the model simulations to understand the vertical transport (after validating the model using GOSAT-SWIR measurements), using GOSAT-TIR measurements we show the seasonal evolution of transport and emissions on the $CH_4$ at different layers of the troposphere using both the model and measurements.

2.   The GOSAT-TIR product shows vivid differences in $CH_4$ from the *a priori* values even in the lower part of the troposphere, where sensitivity of the TANSO-FTS sensor is relatively weak compared to the middle and upper

troposphere. This implies an additional signal of $CH_4$ concentration signal was captured by the TIR observations.

3.   Distinct seasonal variations of $CH_4$ have been observed at the different levels of the troposphere over northern and southern regions of India corresponding to the southwest monsoon (July–September) and early autumn (October–December) seasons. The major contrast between monsoon, and pre- and post-monsoon profiles of $CH_4$ over Indian regions are noticed near the boundary layer levels. This is mainly caused by seasonal change in local emission

strength. Unlike the work by [Guha et al., 2018], we found a strong difference between seasons in the middle and upper troposphere caused by variability in atmospheric circulation and vertical convection.

4.   Even if no averaging kernel incorporated, the mean MIROC4-ACTM and GOSAT-TIR mismatches are within 50 ppb, except for the level of 150 hPa and upward, where the GOSAT-TIR sensitivity becomes very low. Convolution of the modeled profiles with retrieval *a priori* and averaging kernels reduce the mismatch to below uncertainty.





However, the influence of the *a priori* profiles becomes too large with such smoothing. In comparison with AIRS satellite observation [Kavitha and Nair, 2019], finer vertical resolution of GOSAT-TIR allows capturing more detailed features in CH$_4$ vertical profiles. Consequently, we obtained more prominent CH$_4$ patterns related to different regions and seasons.

5.      The significant difference in the methane profile gradient, its seasonal variability (winter and summer) between a priori (derived from the NIES TM simulations) and the MIROC4-ACTM model was revealed in UTLS zone (levels of 150-20 hpa). During monsoon season daily variation in a priori profiles is found in the middle troposphere. Thus, additional studies with use of custom *a priori* profiles in retrieval is of great importance.

6.      Although we found the noticeable error in the model data in phase and amplitude at the end of summer–fall period, the performance of MIROC4-ACTM in CH$_4$ transport in the troposphere and the lower stratosphere was improved due to the use of MIROC4.0 as the meteorological model. Furthermore, an additional analysis with aircraft observations is necessary to analyze the GOSAT-TIR and MIROC4-ACTM mismatch found above the level of 150hPa. Our results suggest that the selection of *a priori* model for satellite data retrieval could play a significant role and should be addressed in the developments of future retrieval systems.

7.      Among the two emission scenarios considered above, the Cao scheme seems to be more balanced than WH for individual regions and the whole South Asia during the monsoon season. In the other periods, no strong difference was found. Using the Cao and WH emission combinations, the annual mean emission for the South Asia region is estimated to $65.7 \pm 2.1$ Tg yr$^{-1}$ for the period 2009-2014.

Overall, the MIROC4-ACTM simulations of CH$_4$ in the Indian regions compare favorably with the GOSAT-TIR samplings, in terms of seasonality and global variability. Inconsistencies seen in the GOSAT-TIR and MIROC4-ACTM comparisons could provide opportunities for further flux optimization with inverse modeling methods. More insight could be obtained after the extension of the released data period of the GOSAT-TIR CH$_4$ product.



*Author contributions.* Conceptualization, methodology: N.S. and P.P.; formal analysis, and original draft preparation: D.B.; GOSAT-TIR CH$_4$ data primary processing: N.S.; MIROC4-ACTM flux optimization: N.C.; discussion, writing
review and editing: all co-authors.

*Competing interests.* The authors declare no conflict of interest.

*Code and data availability.* GOSAT/TANSO-FTS TIR and a priori CH$_4$ data and TIR CH4 averaging kernel data are
provided at http://www.gosat.nies.go.jp/en/. MIROC4-ACTM inversion fluxes are part of the GCP-CH$_4$ database (Saunois et al., ESSD, 2020) and are also downloadable from https://ebcrpa.jamstec.go.jp/~prabir/data/ch4l2r53/gcp2019/. Additional data requests regarding MIROC4-ACTM CH4 concentrations should be addressed to Dmitry Belikov (d.belikov@chiba-u.jp). All data processing codes are developed using Python and can be made available up to request to the corresponding author.


*Acknowledgements.* This research was supported by the Environment Research and Technology Development Fund (2-1802) of the Environmental Restoration and Conservation Agency of Japan. We also acknowledge the MODIS mission scientists and associated NASA personnel for the production of the data used in this research effort. Analyses and visualizations used in this paper were partly produced with the Giovanni online data system, developed and maintained by
the NASA GES DISC.



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
