# Peer review of "Methane vertical profiles over the Indian subcontinent derived from the GOSAT/TANSO-FTS thermal infrared sensor"

_Atmospheric Measurement Techniques, 2020_

## Referee Comment (RC1) · Anonymous Referee #1 · 4 Aug 2020

Review of the manuscript "Methane vertical profiles over. . ." by Belikov et al. submitted to the journal Atmospheric Measurement Techniques

The article "Methane vertical profiles over. . ." by Belikov et al. presents the vertical distribution of methane ($CH_4$) retrieved from the GOSAT TIR measurements focussing on the Indian subcontinent over the period 2009-2014. Coupled to a model (MIROC4), the seasonal variations of $CH_4$ in the lower, middle and upper troposphere are analysed in terms of transport and sources. The sensitivity of the GOSAT $CH_4$ retrievals into the lowermost troposphere is discussed in order to quantify a surface emission of $CH_4$ over India of 51.2 +/- 1.6 Tg yr-1 in 2009-2014.

[Figure]

It is a potentially very interesting paper that may help in the quantification of the CH4 flux over a key source area, namely India and its surroundings. Nevertheless, the article has too many weaknesses to properly address this fundamental issue. In my opinion, the manuscript has a too broad (not precise and/or not rigorous enough) approach of the scientific and technical issues related to CH4 in the observation and in the modelling aspects. These aspects will be dealt in detail below but major ones are: 1) a title that is too vague but consistent with the content of the manuscript, in other words the paper is a mixture of observations, validation, modelling and process studies, but none of them are carefully addressed; 2) the lowermost tropospheric sensitivity of the CH4 observations in the TIR is not satisfactorily addressed/proved since it is too much impacted by the dynamical a priori information used; 3) the 2 model outputs in the lowermost troposphere are compared to observations of GOSAT without using averaging kernels (Fig. 13), consequently no quantitative information can be derived on the CH4 emissions from this. I would propose to the authors to focus on one important aspect of their study, namely the CH4 emissions over India, and to clearly show that GOSAT TIR observations can deal with that. Unfortunately, for all these reasons, I cannot suggest to major revise the manuscript but rather to reject it, and to resubmit a version focussing on one single aspect of their analyses.

MAIN POINTS

1) TITLE & CONTENT

The title, very consistent with the content of the study, is too vague to properly address the rationale and the scientific outcomes of the study. Although the geographical extent of the project is clearly presented, the analysis shows CH4 fields analysed: 1) in three different layers from the lowermost to the uppermost troposphere (800, 500 and 200 hPa), 2) over different time periods: from 2009 to 2014, but focussing on 2011 (with no explanation to why focussing on this particular year) and highlighting climatological fields over 1981-2010 and 2009-2014 underlining the lack of consistency in the data analyses used in the overall study (Figure 2), and 3) combined with 2 model outputs

that are not clearly presented in terms of differences of processes impacting CH4 emissions (section 3.1). This wide range of studies are not rigorously addressed and, by the end of the article, the conclusions are not supported by the presented analyses.

2) LOWERMOST TROPOSPHERE

The analyses of CH4 in the lowermost troposphere are probably the most interesting results presented in the manuscript. Unfortunately, the authors fail to convincingly show that the GOSAT TIR CH4 observations are actually sensitive to 800 hPa. One of the main reasons is the use of an a priori information that is issued from a model, that is to say, that is dynamically evolving in time and space. As a consequence, the CH4 retrievals (whatever the layers considered) are contaminated by this dynamical a priori. Figure 3 left shows typically the obvious relationship between retrieved CH4 and a priori CH4 around 800 hPa (no differences) that is also shown in Figure 3 right where the averaging kernels are peaking at 300 hPa and, for few of them, very difficult to examine since they are labelled in levels and not in pressure, around 500 hPa. Maps of CH4 in the lowermost troposphere at 800 hPa (Figures 4 and 5) also show the strong a priori contamination to the GOSAT CH4 observations over India with almost similar fields in GOSAT and in the a priori CH4, that is not the case at 500 and 200 hPa. The vertical distribution of CH4 (Figures 9 and 10) also highlights this point between GOSAT, a priori and the 2 model outputs at 800 hPa. I would suggest to particularly focus on this issue and carefully show to the reader that GOSAT TIR can actually observe in this layer.

3) CH4 EMISSIONS

The two CH4 emissions used in the article will need to be properly presented and explained in what and why they were employed for the study. Differences in the modelled CH4 fields are clearly shown although some Figures will need to be better presented as for instance Figure 12 that does not present the period over which the data are highlighted. The most important issue in my opinion comes from Figure 13 for which

the modelled CH4 (ACTMCao and ACTMWH) are not convolved with the a priori information using equation (1), that is to say, it is impossible to quantify (as the authors do) the South Asian regional CH4 emission simply based on this Figure since models and observations are not in same space. A more rigorous comparisons will need to be done in order to infer such an important value.

Minor Points

I will not list in detail all the minor points that will need to be improved but only the key ones for a future version.

1) In general, please quantify (absolute and relative values) when making comparisons (differences for instance) and avoid terms as "good agreement". This happens several times in the manuscript.

2) Clearly present which version of the GOSAT CH4 retrieval you have used, whether this version has been validated or not and, above all, if the validation has also focussed on the lowermost troposphere (800 hPa).

3) Explain whether you have used daytime or night-time or both CH4 retrievals since this may impact on all of your results, particularly in the lowermost troposphere.

4) Prophet should be presented in the section "Method" and we should understand why you have used this model. Prior to show the results, you should clearly highlight your methodology to achieve your scientific goal(s).

5) Why did you use so many periods? If important for you, please explain.

6) The Degree of Freedom (DOF) of the CH4 TIR retrievals is "around 1" (L. 192), that is to say, you only have access (in theory) to a columnar information of CH4 from TIR. So, explain how you can still have a sensitivity in the lowermost troposphere with DOF=1.

7) The colour Table (yellow-red) showing the Number of observation points in Figures

4 and 5 does not really highlight the dynamical range over the Indian subcontinent. I would propose to modify it.

8) Some sentences are difficult to understand (English and/or science): L. 204, 243, 257, 260, 356, 366, 413.

9) In general, I would propose a section dedicated to the discussion of the results obtained at least to confront your results to those of other studies.

---

## Author Comment (AC1) · 25 Aug 2020

We thank the reviewer for constructive and helpful suggestions well in advance, and thus allowing us to provide some early responses, which may help us to get further views from the reviewer before the end of the discussion. We appreciate very much this opportunity to discuss online.

Here we would like to share the main directions for improving the article:

- The manuscript structure will be revised as suggested.
- The content will be more adjusted to the AMT requirements.
- To avoid misunderstandings, all considered time periods have been fixed to 2009-2014.
- All requested figures have been replotted using a priori data and the averaging kernels.

The reviewer's specific comments (shown in blue) are addressed below.

**Reply to Anonymous Referee #1 comments**

Review of the manuscript "Methane vertical profiles over…" by Belikov et al. submitted to the journal Atmospheric Measurement Techniques.

The article "Methane vertical profiles over…" by Belikov et al. presents the vertical distribution of methane ($CH_4$) retrieved from the GOSAT TIR measurements focussing on the Indian subcontinent over the period 2009-2014. Coupled to a model (MIROC4), the seasonal variations of $CH_4$ in the lower, middle and upper troposphere are analysed in terms of transport and sources. The sensitivity of the GOSAT $CH_4$ retrievals into the lowermost troposphere is discussed in order to quantify a surface emission of $CH_4$ over India of 51.2 +/- 1.6 Tg yr-1 in 2009-2014.

It is a potentially very interesting paper that may help in the quantification of the $CH_4$ flux over a key source area, namely India and its surroundings. Nevertheless, the article has too many weaknesses to properly address this fundamental issue. In my opinion, the manuscript has a too broad (not precise and/or not rigorous enough) approach of the scientific and technical issues related to $CH_4$ in the observation and in the modelling aspects. These aspects will be dealt in detail below but major ones are: 1) a title that is too vague but consistent with the content of the manuscript, in other words the paper is a mixture of observations, validation, modelling and process studies, but none of them are carefully addressed; 2) the lowermost tropospheric sensitivity of the $CH_4$ observations in the TIR is not satisfactorily addressed/proved since it is too much impacted by the dynamical a priori information used; 3) the 2 model outputs in the lowermost troposphere are compared to observations of GOSAT without using averaging kernels (Fig. 13), consequently no quantitative information can be derived on the $CH_4$ emissions from this. I would propose to the authors to focus on one important aspect of their study, namely the $CH_4$ emissions over India, and to clearly show that GOSAT TIR observations can deal with that. Unfortunately, for all these reasons,

I cannot suggest to major revise the manuscript but rather to reject it, and to resubmit a version focussing on one single aspect of their analyses.

MAIN POINTS

1) TITLE & CONTENT

The title, very consistent with the content of the study, is too vague to properly address the rationale and the scientific outcomes of the study. Although the geographical extent of the project is clearly presented, the analysis shows $CH_4$ fields analysed: 1) in three different layers from the lowermost to the uppermost troposphere (800, 500 and 200 hPa), 2) over different time periods: from 2009 to 2014, but focussing on 2011 (with no explanation to why focussing on this particular year) and highlighting climatological fields over 1981-2010 and 2009-2014 underlining the lack of consistency in the data analyses used in the overall study (Figure 2), and 3) combined with 2 model outputs that are not clearly presented in terms of differences of processes impacting $CH_4$ emissions (section 3.1). This wide range of studies are not rigorously addressed and, by the end of the article, the conclusions are not supported by the presented analyses.

We agree, the topic of this study is quite broad due to the complexity of the research topic and several data sets used in the analysis. To meet the requirements of AMT, we decided to narrow the range of studies to GOSAT-TIR specific subjects mainly. As a result, the title will be reformulated as "Interpretation of GOSAT CH4 vertical profiles over the Indian subcontinent coupled with MIROC4-ACTM model".

As stated in the manuscript "This study uses the GOSAT-TIR $CH_4$ product (version V1), which is released for the period from April 23, 2009, through May 24, 2014". We also focused on 2011 to plot seasonal distributions of specific parameters on lat-lon grid (Fig. 3-4, 7-8), as this year has fewer gaps in GOSAT-TIR observations. For meteorological fields depicted in Fig. 2 climatological values were replaced with ones for 2011. To address time variability time series and time-altitude cross-sections were used (Fig. 11, 13).

The used 2 two model outputs are related to $CH_4$ emissions from wetland as described in two different schemes developed by Cao et al. [1996] and Walter et al. [2001]. The main difference between these approaches in most cases can be explained by the combined effect of changes in soil temperature and the position of the water table. The dependence of $CH_4$ emissions on water table depth is complicated and highly non-linear, so the strength of the fluxes is mostly variable in lowland areas along large rivers. In general, the WH scheme fluxes are about 5-10% larger the Cao, excepting the WIGP, EIGP, and NEI regions of India and Bangladesh where the maximum difference reaches 20-40% (Fig. 6). Besides, there are small hot spots in Southeast Asia (e.g. Mekong River Delta).

**2) LOWERMOST TROPOSPHERE**

The analyses of CH₄ in the lowermost troposphere are probably the most interesting results presented in the manuscript. Unfortunately, the authors fail to convincingly show that the GOSAT TIR CH₄ observations are actually sensitive to 800 hPa. One of the main reasons is the use of an a priori information that is issued from a model, that is to say, that is dynamically evolving in time and space. As a consequence, the CH₄ retrievals (whatever the layers considered) are contaminated by this dynamical a priori. Figure 3 left shows typically the obvious relationship between retrieved CH₄ and a priori CH₄ around 800 hPa (no differences) that is also shown in Figure 3 right where the averaging kernels are peaking at 300 hPa and, for few of them, very difficult to examine since they are labelled in levels and not in pressure, around 500 hPa. Maps of CH₄ in the lowermost troposphere at 800 hPa (Figures 4 and 5) also show the strong a priori contamination to the GOSAT CH₄ observations over India with almost similar fields in GOSAT and in the a priori CH₄, that is not the case at 500 and 200 hPa. The vertical distribution of CH₄ (Figures 9 and 10) also highlights this point between GOSAT, a priori and the 2 model outputs at 800 hPa. I would suggest to particularly focus on this issue and carefully show to the reader that GOSAT TIR can actually observe in this layer.

For easy examination, labels of figure 3 (right panels) was updated to show pressure rather than levels.

[Figure]

New figure 3: Seasonal mean (July-September 2011) over regions of Southern India (upper panel), and Northeast India (bottom panel). The left columns show the GOSAT-TIR CH₄ *a priori* with 1-σ STD uncertainty of the mean (red line with error bars) and GOSAT-TIR CH₄ profile with the retrieval error (blue line with the shaded area); the right columns depict the averaging kernels of GOSAT-TIR CH₄

retrievals averaged over time. There are 22 lines for GOSAT-TIR retrievals, corresponding to the retrieval layers used in each of them.

Following the recommendations of the reviewer and the requirements of the AMT journal, the main research will focus on GOSAT-TIR observations in the middle and upper part of the atmosphere (500-300 hPa), where the sensitivity of the TIR instrument is relatively high. For this the corresponding figures have been updated using a priori data and the averaging kernels.

[Figure]

New figure 4: Latitude-longitude distributions of $CH_4$ at the levels of 800, 500, and 300hPa (the left, middle, and right panels respectively) observed by GOSAT-TIR for the season AMJ 2011. The first upper panels (a1-a3) show GOSAT-TIR $CH_4$, the second upper panels (b1-b3) show GOSAT-TIR a priori $CH_4$, the third upper panels (c1-c3) show difference between the GOSAT-TIR observed and a priori distributions, respectively.

**3) $CH_4$ EMISSIONS**

The two $CH_4$ emissions used in the article will need to be properly presented and explained in what and why they were employed for the study. Differences in the modelled $CH_4$ fields are clearly shown although some Figures will need to be better presented as for instance Figure 12 that does not present the period over which the data are highlighted. The most important issue in my opinion

comes from Figure 13 for which the modelled CH₄ (ACTMCao and ACTMWH) are not convolved with the a priori information using equation (1), that is to say, it is impossible to quantify (as the authors do) the South Asian regional CH₄ emission simply based on this Figure since models and observations are not in same space. A more rigorous comparisons will need to be done in order to infer such an important value.

Figure 12 shows multi-year (2009-2014) seasonal variation of CH₄ from GOSAT-TIR and MIROC4-ACTM. To avoid misunderstanding we updated this information to the caption. The plot was also updated.

[Figure]

New figure 12: Multi-year (2009-2014) seasonal variation of CH₄ (right y-axis) derived by the implementation of the Prophet model for levels of 800 (red lines) and 300 hPa (blue lines) over considered regions from GOSAT-TIR (solid line), $\mathbf{ACTM_{Cao}^{AK}}$ (dashed line), and $\mathbf{ACTM_{WH}^{AK}}$ (dotted line),

respectively. The background bar plots represent Cao (dark grey) and WH (light grey) CH$_4$ fluxes (left y-axis), respectively. Please note the different scale of y-axes (left) for fluxes.

Figure 13 was replotted using AK and a priori information using equation (1).

[Figure]

New figure 13: Time series of CH$_4$ averaged over the area of South Asia for levels of a) 300, b) 500 and c) 800 hPa respectively. Symbols state the GOSAT-TIR observations, red and blue lines are for **ACTM$_{Cao}^{AK}$**, and**ACTM$_{WH}^{AK}$**, respectively. Solid and dashed lines are for monthly and yearly averaged concentrations (left y-axis), the dotted line shows the difference between the model simulations (right y-axis), respectively.

Minor Points

I will not list in detail all the minor points that will need to be improved but only the key ones for a future version.

1) In general, please quantify (absolute and relative values) when making comparisons (differences for instance) and avoid terms as "good agreement". This happens several times in the manuscript.

The corresponding sentences will be revised.

2) Clearly present which version of the GOSAT CH$_4$ retrieval you have used, whether this version has been validated or not and, above all, if the validation has also focussed on the lowermost troposphere (800 hPa).

Agree. Due to technical issues in the typeset process, the version mark (V1) disappeared in several places. Thus, the version was added one more time, for example: "This study uses the GOSAT-TIR CH$_4$ product (version V1), which is released for the period from April 23, 2009, through May 24, 2014."

This version was validated in several recent works [Holl et al., 2016; Zou et al., 2016; Olsen et al. 2017], as stated in the Introduction. However, the lowermost troposphere (800 hPa) level is not covered in those works.

3) Explain whether you have used daytime or night-time or both CH4 retrievals since this may impact on all of your results, particularly in the lowermost troposphere.

The following sentence will be added: "GOSAT measures in TIR spectra twice a day. The first one is performed along with the SWIR band at local noon ±1 h, the second one near midnight (local noon ±12 hr). Daytime observations are being screened more thoroughly using the Cloud and Aerosol Imager (CIA). At night, filtering is carried out only by the TIR algorithm. Therefore, the number of scenes of daytime and night-time observations can differ significantly, especially during the monsoon period. However, the average difference between daytime and night-time observed CH$_4$ profiles is usually within ±20 ppb. We leave a more detailed analysis of the daily variation of methane outside the scope of this work."

4) Prophet should be presented in the section "Method" and we should understand why you have used this model. Prior to show the results, you should clearly highlight your methodology to achieve your scientific goal(s).

The Prophet section was moved to "Method":

2.7 The Prophet analysis and forecasting model

GOSAT-TIR CH$_4$ observations show large temporal and spatial heterogeneity, which complicate the deriving of seasonal cycle variation. To derive cyclic variations from noisy and time irregular datasets use of special methods is highly desirable. The Prophet time-series analysis and forecasting model [Taylor and Letham, 2018] was implemented to derive multi-year (2009-2014) seasonal variation of CH4 from GOSAT-TIR and MIROC4-ACTM. The Prophet is a novel analysis and forecasting model, which performs smoothing of time-series data based on a generalized additive model with three main components: trend, seasonality, and holidays. Compared to traditional exponential smoothing, the Prophet can easily handle temporal patterns with multiple periods and has no requirements

regarding the regularity of measurement spacing. The model has a robust performance in the presence of missing data and trend shifts and typically handles outliers well while working with time-series that have several seasons of historical data with strong seasonal patterns. The Prophet manages the use of all data points for the study period, thereby increasing accuracy and reducing sensitivity to random outliers [Belikov et al. 2019].

Later in the text:

3.5 Seasonal variation of $CH_4$

The Prophet time-series analysis and forecasting model [Taylor and Letham, 2018] was implemented to derive the mean seasonal cycle of CH4 from GOSAT-TIR and MIROC4-ACTM for the levels of 800 and 300 hPa for 2009-2014 ignoring year-to-year variations (Fig. 13).

5) Why did you use so many periods? If important for you, please explain.

To avoid misunderstandings all considered time periods were revised, as stated above.

6) The Degree of Freedom (DOF) of the $CH_4$ TIR retrievals is "around 1" (L. 192), that is to say, you only have access (in theory) to a columnar information of $CH_4$ from TIR. So, explain how you can still have a sensitivity in the lowermost troposphere with DOF=1.

DOF means the number of purely (mathematically) independent pieces of information. The real atmospheric layers correlate with each other, so even if its DOF is one, TIR has some sensitivity to lower tropospheric CH4 concentration judging from small differences seen between TIR and a priori CH4 concentrations in the lower troposphere. However, the sensitivity to the lower troposphere is smaller than the upper levels and we should not treat TIR CH4 data in the lower troposphere in a similar manner to those in the upper levels; therefore, we more focus on the upper troposphere where TIR measurements has the most sensitivity in the revised manuscript.

7) The colour Table (yellow-red) showing the Number of observation points in Figures 4 and 5 does not really highlight the dynamical range over the Indian subcontinent. I would propose to modify it.

The colour map of the figure was modified.

[Figure]

New figure 4: Latitude-longitude distributions of GOSAT-TIR $CH_4$ observation points numbers at the levels of 800, 500, and 300hPa (the left, middle, and right panels respectively) for the season AMJ 2011.

8) Some sentences are difficult to understand (English and/or science): L. 204, 243, 257, 260, 356, 366, 413.

Will be revised

9) In general, I would propose a section dedicated to the discussion of the results obtained at least to confront your results to those of other studies.

Agree. The discussion section will be added.

---

## Referee Comment (RC2) · Anonymous Referee #2 · 13 Oct 2020

General Comments:

This manuscript presents the vertical distribution of CH4 from GOSAT retrievals over India within the context of elucidating: a) issues related to GOSAT sensitivities and 'a priori' profiles, b) processes influencing the spatiotemporal distribution (emissions, transport), and c) variability across the region. All these aspects are relevant to atmospheric CH4 investigations especially that there are only few retrievals (and less in-situ datasets) available. These three aspects are also described in the paper within a fairly reasonable depth. However, I have two major concerns, which require attention from the authors. That is,

1) The relevance of this study to the scope of AMT is unclear. Unless the paper is refocused on issues with GOSAT retrievals esp the choice of a priori and/or highlighting the sensitivities of GOSAT and the proper use and interpretation of these retrievals. The paper already presented several figures and discussion to these points but more emphasis could be made to bring it closer to the scope of AMT.

2) Lacks comparison (verification) with available independent measurements. While it is understandable that there are only few measurements available, model-based comparisons are not sufficient. Some efforts to compare with other measurements (aircraft or ground based or other retrievals from different instrument) would strengthen the paper's findings.

Specific Comments: 1) Title is a bit misleading as the paper does not discuss this in depth.

2) Abstract states that the objective is to understand retrieval sensitivity, but the results are more towards comparison of CH4 variations across with models including emissions, without any independent measurements to compare with.

3) Line 16: Stating "22 vertical levels . . . provide critical information' is misleading. Might be better to state its DOFS and vertical sensitivities.

4) Line 18: 'excepting' ?

5) Line 95-100. It would be great to describe the retrieval algorithm including a priori error covariance assumptions (if this is an optimal estimation). A short description as well of NEIS relative to MIROC (esp emissions used in NEIS).

6) Line 105-110. This is a very useful discussion of GOSAT retrievals. Why are these other retrievals not used for comparison over India in this study?

7) Section 3.2. This is also a very useful section. If DOFS is 1, why do we have profile information?

8) Line 213. 'resampled' ?

9) Line 221. It may be interesting to show differences in AK over land and ocean.

10) Figure 4 & 5. These figures are informative.

11) Figure 9, Line 284-287. What about the retrieval errors (from the a posteriori estimates)? Please elaborate 'we found that differences between a priori and retrieved CH4 profiles are larger than its retrieval error...'.

12) Line 311-312. This sentence is unclear. Please restate.

13) Line 316-317. Is this a study where different a priori profiles (and I assumed the error covariance is the same) are used in the retrievals. Please make sure the use of 'a priori' is consistent across the manuscript (including italics and non-italics).

14) Line 320-324. This is a useful discussion and should be highlighted more.

15) Figure 11. More discussion on this (relative to a priori) would strengthen this paper.

16) Section 3.5. This looks like more of a comparison with ACTM and elucidating differences. It may be better if this can be made a separate section with slightly different heading.

17) Line 375. 'ACTM WH is superior to ACTM CAO'. It's unclear from the bar graphs.

18) Line 399. What is the basis for 10-15 Tg yr overestimation (How was this number derived?)

19) Conclusions. While comparison with models is informative, it remains to be proven if the differences between GOSAT and modeled profiles reflect 'real' differences — unless independent measurements (and/or retrieval experiments) are made.

---

## Author Comment (AC2) · 10 Nov 2020

We thank the reviewer for constructive and helpful suggestions well in advance, and thus allowing us to provide some early responses, which may help us to get further views from the reviewer before the end of the discussion. We appreciate very much this opportunity to discuss online.

Here we would like to share the main directions for improving the article:

- The manuscript structure was revised as suggested
- The content was more adjusted to the AMT requirements
- All requested figures have been replotted

The reviewer's specific comments (shown in blue) are addressed below.

**Reply to Anonymous Referee #2 comments**

General Comments:

This manuscript presents the vertical distribution of CH4 from GOSAT retrievals over India within the context of elucidating: a) issues related to GOSAT sensitivities and 'a priori' profiles, b) processes influencing the spatiotemporal distribution (emissions, transport), and c) variability across the region. All these aspects are relevant to atmospheric CH4 investigations especially that there are only few retrievals (and less in-situ datasets) available. These three aspects are also described in the paper within a fairly reasonable depth. However, I have two major concerns, which require attention from the authors. That is,

1) The relevance of this study to the scope of AMT is unclear. Unless the paper is refocused on issues with GOSAT retrievals esp the choice of a priori and/or highlighting the sensitivities of GOSAT and the proper use and interpretation of these retrievals. The paper already presented several figures and discussion to these points but more emphasis could be made to bring it closer to the scope of AMT.

We agree the topic of this study is quite broad due to the complexity of the research topic and several data sets used in the analysis. To meet the requirements of AMT, we decided to narrow the range of studies to GOSAT-TIR specific subjects mainly.

2) Lacks comparison (verification) with available independent measurements. While it is understandable that there are only few measurements available, model-based comparisons are not sufficient. Some efforts to compare with other measurements (aircraft or ground based or other retrievals from different instrument) would strengthen the paper's findings.

As shown above, this issue is described in Discussion L.416-426: "Despite essential progress, the development of satellite methods for studying atmospheric methane is obstructed by a number of limitations. The launch rate of new orbital instruments is significantly ahead of the development of

a ground-based and aircraft measurement network for their validation. Due to the scarcity of suitable aircraft observations over India, validation of GOSAT-TIR profiles cannot cover a variety of seasons and land regions studied in this work. However, in the newly prepared paper by N. Saitoh "Intensive validation analysis of GOSAT/TANSO-FTS thermal infrared $CH_4$ data (version 1) based on aircraft observations" (to be submitted to "Remote Sensing"), the intensive validation work of GOSAT-TIR $CH_4$ profiles is described. In this paper, global comparisons are conducted based on HIPPO, CARIBIC, JMA, and CONTRAIL/ASE aircraft observations. In low latitudes corresponding to the India location, compared datasets include CARIBIC profiles over MAA (Chennai, India), BOG (El Dorado, Colombia), and CCS (Venezuela) airports and CONTRAIL/ASE over GUAM (US) airport. The validations show that TIR V1 $CH_4$ and aircraft $CH_4$ profiles agreed with each other within 10-15 ppb and there was no evident seasonal dependence in the $CH_4$ differences."

**Specific Comments:**

1) Title is a bit misleading as the paper does not discuss this in depth.

We agree. The title was revised as "Interpretation of GOSAT CH4 vertical profiles over the Indian subcontinent: effect of a priori and averaging kernels".

2) Abstract states that the objective is to understand retrieval sensitivity, but the results are more towards comparison of CH4 variations across with models including emissions, without any independent measurements to compare with.

The text of the paper was seriously elaborated to meet this requirement.

3) Line 16: Stating "22 vertical levels … provide critical information' is misleading. Might be better to state its DOFS and vertical sensitivities.

Agree. Revised as follows L.15-17: "A comparison of modeled and retrieved CH4 vertical profiles shows the GOSAT/TANSO-FTS TIR sensitivity is sufficient to provide critical information about transport from the top of the boundary layer to the upper troposphere and lower stratosphere in a consistent manner."

4) Line 18: 'excepting' ?

Revised as L.18-19: "...50 ppb, except for the altitude range above 150 hPa, where…"

5) Line 95-100. It would be great to describe the retrieval algorithm including a priori error covariance assumptions (if this is an optimal estimation). A short description as well of NEIS relative to MIROC (esp emissions used in NEIS).

The retrieval algorithm description was updated L.96-99: "The retrieval algorithm for the TANSO-FTS TIR V1 $CH_4$ product is basically the same as for the V1 $CO_2$ described in Saitoh et al. (2016). It adopted a nonlinear maximum a posteriori (MAP) method with linear mapping. *A priori* covariance matrix for $CH_4$ in the V1 $CH_4$ retrieval is set to be a diagonal matrix with vertically fixed diagonal elements with a standard deviation of 4%."

NIES simulation setup was updated L.100-106: "For simulation NIES TM used the monthly varying flux for 2000 (575 Tg yr$^{-1}$) based on the Emission Database for Global Atmospheric Research (EDGAR) version 32FT2000 (Olivier and Berdowski, 2001) for anthropogenic CH4, and on GISS emissions (Fung et al., 1991) for natural CH4, as obtained from Patra et al. (2009). The chemical destruction of CH4 by OH radicals was calculated based on climatological monthly mean OH radical concentrations (Spivakovsky et al., 2000) and a temperature-dependent rate constant."

6) Line 105-110. This is a very useful discussion of GOSAT retrievals. Why are these other retrievals not used for comparison over India in this study?

We think "these other retrievals" mean other $CH_4$ data from AIRS, ACE-FTS, and MIPAS. ACE-FTS and MIPAS are solar-occultation and limb-viewing sensors, respectively, so their horizontal resolutions are too law and their measurements are too sparse to discuss detailed features of $CH_4$ over India. AIRS has a much lower spectral resolution than GOSAT, so GOSAT is more suitable for the discussions of $CH_4$ vertical distributions over India.

7) Section 3.2. This is also a very useful section. If DOFS is 1, why do we have profile information? DOF means the number of purely (mathematically) independent pieces of information. The real atmospheric layers correlate with each other, so even if its DOF is one, TIR has some sensitivity to lower tropospheric CH4 concentration judging from small differences seen between TIR and a priori CH4 concentrations in the lower troposphere. However, the sensitivity to the lower troposphere is smaller than the upper levels and we should not treat TIR CH4 data in the lower troposphere in a similar manner to those in the upper levels; therefore, we more focus on the upper troposphere where TIR measurements has the most sensitivity in the revised manuscript.

Added to the text L.135-138:" The degrees of freedom (DOF) of signal for $CH_4$ observation by GOSAT-TIR band (V1 algorithm version) is around 1 over low-latitude part of India. DOF means the number of purely (mathematically) independent piece of information. However, the real atmospheric layers correlate with each other, so even if its DOF is close to 1, TIR has ability to derive new knowledge about $CH_4$ concentration."

8) Line 213. 'resampled' ?

'resampled' -> regridded

9) Line 221. It may be interesting to show differences in AK over land and ocean.

[Figure]

Updated fig 1 include AK for two land and one oceanic regions as show above.

10) Figure 4 & 5. These figures are informative.

Agree. Thank you

11) Figure 9, Line 284-287. What about the retrieval errors (from the a posteriori estimates)? Please elaborate 'we found that differences between a priori and retrieved CH4 profiles are larger than its retrieval error…'.

This sentence revised L.309-314: "The variation of GOSAT-TIR sensitivity are taking into account by the implementation of the *a priori* profiles and AK functions (Fig. 1) to the modelled data sets ($ACTM_{Cao}^{AK}$ and $ACTM_{WH}^{AK}$). The variabilities shown in Figure 10 are larger than GOSAT-TIR retrieval errors, derived here as the diagonal elements of the posteriori error covariance matrices based on the MAP method, which include random error components of the retrieval. Therefore, GOSAT-TIR and model show good agreements (mismatch is inside 1-$\sigma$ STD uncertainty) within both errors (natural variabilities and retrieval random errors)."

12) Line 311-312. This sentence is unclear. Please restate.

This sentence revised L.342-346: "Fig. 11 shows the use of GOSAT-TIR AK functions have significant smoothing effect, approaching the MIROC4-ACTM model profiles to *a priori* so much that the difference between the calculations for the Cao and WH emission scenarios becomes barely distinguishable. This is especially visible above the level of 150 hPa, where the sensitivity of GOSAT-TIR there drops sharply and the satellite retrievals and the AK convolved model profiles strongly follow the *a priori* profiles.

"

13) Line 316-317. Is this a study where different a priori profiles (and I assumed the error covariance is the same) are used in the retrievals. Please make sure the use of 'a priori' is consistent across the manuscript (including italics and non-italics).

Revised through the text

14) Line 320-324. This is a useful discussion and should be highlighted more.

Discussion of *a priori* profiles was extended L.432-456:

"*A priori* profiles play an essential role in processing a satellite signal, especially for the $CH_4$, which has a significant change in a lifetime with altitude. The choice of such a profile (usually provided by model calculations) is a critical point since due to small DOF the retrieval algorism cannot overcome large errors in input data. The TransCom-$CH_4$ experiment [Patra et al., 2011] showed a significant scatter between the participated models, including the NIES model later selected for calculating

GOSAT-TIR *a priori* profile (described in 2.2). In this study, the difference in the methane profile gradient, its seasonal variability (winter and summer) between *a priori* and the MIROC4-ACTM model was revealed in the UTLS zone (levels of 150-20 hPa). Apparently, the difference in the modelling of UTLS is a key factor, as the MIROC4-ACTM meteorological parameter is driven by recently updated reanalysis and its vertical resolution (67 sigma-pressure levels) is quite higher than that of the NIES (47 sigma levels). Even more important, the stratospheric part of the NIES model was adjusted to the observed age of air for CO2 and long-term satellite observations from HALOE for $CH_4$ [Saeki et al. 2013]. Therefore, the reason for the misfits (GOSAT-TIR *a priori* vs ACTM) extremely controversial without additional studies with the use of custom *a priori* profiles in retrieval.

In the case of long-term projects, the updating of *a priori* data in accordance with the current progress becomes important. From the moment of the GOSAT launch, the calculation of *a prior* profiles is carried out according to the same scheme. This is important for the long-term consistency of the GOSAT-TIR $CH_4$ product but does not take into account the recent improvements (e.g. new reanalysis data, higher vertical resolution, and convective parameterizations) implemented in MIROC4-ACTM [Patra et al., 2018], as well as further understanding of the $CH_4$ budget [Saunois et al., 2020].

As the release of a new version of retrieval algorithms designed to improve and update satellite products is not regular, perhaps, for further progress, the retrieval process should be open access with a possibility of use of custom *a priori* information. Therefore, this will require greater transparency of technical information from satellite projects and significant optimization of retrieval calculations, since such tasks are requiring large computational resources. Promising is performing of retrieval inter-comparison projects (one algorithm with a variety of *a priori* information and set of algorithms with the same of *a priori*), as it was done for CTMs development [Patra et al., 2011]. However, this *a priori* profile issue remains hidden (but no less relevant) in the field of the main efforts of the scientific community working with the column-averaged burden ($XCH_4$) derived from the SWIR band."

15) Figure 11. More discussion on this (relative to a priori) would strengthen this paper.

The following sentences are added to L.361-366: "Apparently, the *a priori* profiles from NIES model shows insufficient vertical transport due to incomplete convective parameterization required to simulate tracer transport under monsoon conditions (Fig. 12). This problem forced the transition to a more sophisticated reanalysis JRA-25/JCDAS (Japanese 25-yr Reanalysis/Climate Data

Assimilation System developed by the Japan Meteorological Agency (JMA)) and the adaptation of the new parameterization as described by Belikov et al. [2013]. The GOSAT-TIR retrieval is trying to compensate for such a concentration deficit in the upper troposphere. On the other hand, the overestimation by MIROC4-ACTM despite the implemented modifications [Patra et al., 2018] is also possible."

16) Section 3.5. This looks like more of a comparison with ACTM and elucidating differences. It may be better if this can be made a separate section with slightly different heading.
Section 3.5 was spited into 2: 3.5 Seasonal variation of $CH_4$ and 3.6 Regional CH4 emission estimation

17) Line 375. 'ACTM WH is superior to ACTM CAO'. It's unclear from the bar graphs.
This sentence is related to new Fig.13 now, where $ACTM_{WH}$ (blue line) is higher than $ACTM_{Cao}$ (red line) for all 3 levels (panels a-c). For more details, the simulation difference ($ACTM_{WH}$ - $ACTM_{Cao}$) is shown by dashed line.

18) Line 399. What is the basis for 10-15 Tg yr overestimation (How was this number derived?)
Revised text added to L.469-478: "The problem of reliable data lack for estimation regional $CH_4$ budgets can be mitigated by GOSAT-TIR $CH_4$. Relying on our comparison (Fig. 14), we suggest the Cao flux combination with the annual mean emission of 65.7 ± 5.8 Tg yr$^{-1}$ for the period 2009-2014 as more plausible. This confirms the assessment made by Patra et al. [2016], indicating that the EDGAR inventory (version 4.2FT2010) with a value of 73.3–83.2 Tg yr$^{-1}$ overestimated the South Asia regional emission by 10-15 Tg yr$^{-1}$. A significant part of the extra fluxes is concentrated in a few relatively small regions in the Northen India (fig. 2). However, our best estimate emission of 51.2 ± 4.6 Tg yr$^{-1}$ over India is much greater than 19.6–24.3 Tg yr$^{-1}$ estimated by [Ganesan et al., 2017], combined *in situ* data of different time coverage and SWIR $CH_4$ retrievals in the trajectory-based modelling framework. Simulation with two scenarios showed that during the monsoon significant $CH_4$ amount due to extra fluxes can fast propagated to the UTLS zone and not been detected by ground-based measurements. This emphasizes the importance of correctly accounting for the effects of vertical transport for emission estimating."

19) Conclusions. While comparison with models is informative, it remains to be proven if the differences between GOSAT and modeled profiles reflect 'real' differences — unless independent measurements (and/or retrieval experiments) are made.

As shown above, this issue is described in Discussion L.416-426.

---

## Author Comment (AC3) · 10 Nov 2020

This version has been updated in accordance with the comments by Anonymous Referee #2 and the final version of the article.

We thank the reviewer for constructive and helpful suggestions well in advance, and thus allowing us to provide some early responses, which may help us to get further views from the reviewer before the end of the discussion. We appreciate very much this opportunity to discuss online.

Here we would like to share the main directions for improving the article:

- The manuscript structure was revised as suggested.
- The content was more adjusted to the AMT requirements.
- To avoid misunderstandings, all considered time periods have been fixed to 2009-2014.
- All requested figures have been replotted using a priori data and the averaging kernels.

The reviewer's specific comments (shown in blue) are addressed below.

**Reply to Anonymous Referee #1 comments**

Review of the manuscript "Methane vertical profiles over..." by Belikov et al. submitted to the journal Atmospheric Measurement Techniques.

The article "Methane vertical profiles over..." by Belikov et al. presents the vertical distribution of methane (CH4) retrieved from the GOSAT TIR measurements focussing on the Indian subcontinent over the period 2009-2014. Coupled to a model (MIROC4), the seasonal variations of CH4 in the lower, middle and upper troposphere are analysed in terms of transport and sources. The sensitivity of the GOSAT CH4 retrievals into the lowermost troposphere is discussed in order to quantify a surface emission of CH4 over India of 51.2 +/- 1.6 Tg yr-1 in 2009-2014.

It is a potentially very interesting paper that may help in the quantification of the CH4 flux over a key source area, namely India and its surroundings. Nevertheless, the article has too many weaknesses to properly address this fundamental issue. In my opinion, the manuscript has a too broad (not precise and/or not rigorous enough) approach of the scientific and technical issues related to CH4 in the observation and in the modelling aspects. These aspects will be dealt in detail below but major ones are: 1) a title that is too vague but consistent with the content of the manuscript, in other words the paper is a mixture of observations, validation, modelling and process studies, but none of them are carefully addressed; 2) the lowermost tropospheric sensitivity of the CH4 observations in the TIR is not satisfactorily addressed/proved since it is too much impacted by the dynamical a priori information used; 3) the 2 model outputs in the lowermost troposphere are compared to observations of GOSAT without using averaging kernels (Fig. 13), consequently no

quantitative information can be derived on the CH4 emissions from this. I would propose to the authors to focus on one important aspect of their study, namely the CH4 emissions over India, and to clearly show that GOSAT TIR observations can deal with that. Unfortunately, for all these reasons, I cannot suggest to major revise the manuscript but rather to reject it, and to resubmit a version focussing on one single aspect of their analyses.

**MAIN POINTS**

**1) TITLE & CONTENT**

The title, very consistent with the content of the study, is too vague to properly address the rationale and the scientific outcomes of the study. Although the geographical extent of the project is clearly presented, the analysis shows CH4 fields analysed: 1) in three different layers from the lowermost to the uppermost troposphere (800, 500 and 200 hPa), 2) over different time periods: from 2009 to 2014, but focussing on 2011 (with no explanation to why focussing on this particular year) and highlighting climatological fields over 1981-2010 and 2009-2014 underlining the lack of consistency in the data analyses used in the overall study (Figure 2), and 3) combined with 2 model outputs that are not clearly presented in terms of differences of processes impacting CH4 emissions (section 3.1). This wide range of studies are not rigorously addressed and, by the end of the article, the conclusions are not supported by the presented analyses.

We agree, the topic of this study is quite broad due to the complexity of the research topic and several data sets used in the analysis. To meet the requirements of AMT, we decided to narrow the range of studies to GOSAT-TIR specific subjects mainly. As a result, the title was reformulated as "Interpretation of GOSAT CH4 vertical profiles over the Indian subcontinent: effect of a priori and averaging kernels".

As stated in the manuscript "This study uses the GOSAT-TIR CH4 product (version V1), which is released for the period from April 23, 2009, through May 24, 2014". We also focused on 2011 to plot seasonal distributions of specific parameters on lat-lon grid (Fig. 5-9), as this year has fewer gaps in GOSAT-TIR observations. For meteorological fields depicted in Fig. 4 climatological values were replaced with ones for 2011. To address time variability time series and time-altitude cross-sections were used (Fig. 12, 14).

The used two model outputs are related to CH4 emissions from wetland as described in two different schemes developed by Cao et al. [1996] and Walter et al. [2001]. The main difference between these approaches in most cases can be explained by the combined effect of changes in soil temperature and the position of the water table. The dependence of CH4 emissions on water table depth is complicated and highly non-linear, so the strength of the fluxes is mostly variable in lowland areas along large rivers. In general, the WH scheme fluxes are about 5-10% larger the Cao, excepting

the WIGP, EIGP, and NEI regions of India and Bangladesh where the maximum difference reaches 20-40% (Fig. 2). Besides, there are small hot spots in Southeast Asia (e.g. Mekong River Delta).

**2) LOWERMOST TROPOSPHERE**

The analyses of CH4 in the lowermost troposphere are probably the most interesting results presented in the manuscript. Unfortunately, the authors fail to convincingly show that the GOSAT TIR CH4 observations are actually sensitive to 800 hPa. One of the main reasons is the use of an a priori information that is issued from a model, that is to say, that is dynamically evolving in time and space. As a consequence, the CH4 retrievals (whatever the layers considered) are contaminated by this dynamical a priori. Figure 3 left shows typically the obvious relationship between retrieved CH4 and a priori CH4 around 800 hPa (no differences) that is also shown in Figure 3 right where the averaging kernels are peaking at 300 hPa and, for few of them, very difficult to examine since they are labelled in levels and not in pressure, around 500 hPa. Maps of CH4 in the lowermost troposphere at 800 hPa (Figures 4 and 5) also show the strong a priori contamination to the GOSAT CH4 observations over India with almost similar fields in GOSAT and in the a priori CH4, that is not the case at 500 and 200 hPa. The vertical distribution of CH4 (Figures 9 and 10) also highlights this point between GOSAT, a priori and the 2 model outputs at 800 hPa. I would suggest to particularly focus on this issue and carefully show to the reader that GOSAT TIR can actually observe in this layer.

For easy examination, labels of figure 1 (right panels) was updated to show pressure rather than levels.

New figure 3 (Now figure 1): Seasonal mean (July-September 2011) over regions of Southern India (upper panel), Northeast India (middle panel), and Arabian Sea (bottom panel). The left columns show the GOSAT-TIR CH4 *a priori* with 1- $\sigma$  STD uncertainty (red line with error bars) and GOSAT-TIR CH4 profile with the retrieval error (blue line with shaded area); the right columns depict AK matrix of GOSAT-TIR CH4 retrievals averaged over time. There are 22 lines for GOSAT-TIR retrievals, corresponding to the retrieval layers used in each of them.

Following the recommendations of the reviewer and the requirements of the AMT journal, the main research focused on GOSAT-TIR observations in the middle and upper part of the atmosphere (500-300 hPa), where the sensitivity of the TIR instrument is relatively high. For this the corresponding figures have been updated using a priori data and the averaging kernels.